# When Less Language is More: Language-Reasoning Disentanglement Makes LLMs Better Multilingual Reasoners

**Weixiang Zhao[1]\*, Jiahe Guo[1]\*, Yang Deng[2], Tongtong Wu[3], Wenxuan Zhang[4], Yulin Hu[1]**
**Xingyu Sui[1], Yanyan Zhao[1]†, Wanxiang Che[1], Bing Qin[1], Tat-Seng Chua[5], Ting Liu[1]**
[1]Harbin Institute of Technology, [2]Singapore Management University, [3]Monash University
[4]Singapore University of Technology and Design, [5]National University of Singapore
{wxzhao,jhguo,yyzhao}@ir.hit.edu.cn

## Abstract

Multilingual reasoning remains a significant challenge for large language models (LLMs), with performance disproportionately favoring high-resource languages. Drawing inspiration from cognitive neuroscience, which suggests that human reasoning functions largely independently of language processing, we hypothesize that LLMs similarly encode reasoning and language as separable components that can be disentangled to enhance multilingual reasoning. To evaluate this, we perform a causal intervention by ablating language-specific representations at inference time. Experiments on 10 open-weight LLMs spanning 11 typologically diverse languages show that this language-specific ablation consistently boosts multilingual reasoning performance. Layer-wise analyses further confirm that language and reasoning representations can be effectively disentangled throughout the model, yielding improved multilingual reasoning capabilities, while preserving top-layer language features remains essential for maintaining linguistic fidelity. Compared to post-training methods such as supervised fine-tuning or reinforcement learning, our training-free language-reasoning disentanglement achieves comparable or superior results with minimal computational overhead. These findings shed light on the internal mechanisms underlying multilingual reasoning in LLMs and suggest a lightweight and interpretable strategy for improving cross-lingual generalization. Our code is available at: https://github.com/MuyuenLP/Language-Reasoning-Disentangle.

## 1 Introduction

Recent advances in reasoning large language models—as exemplified by OpenAI's o1/o3/o4 models [Jaech et al., 2024, OpenAI, 2025] and the DeepSeek-R1 series [Guo et al., 2025]—have significantly advanced the capacity of language models to handle complex reasoning tasks. These improvements stem from multi-stage post-training pipelines, notably supervised fine-tuning and large-scale reinforcement learning with reasoning-enhanced objectives [Guo et al., 2025, Kimi et al., 2025, Qwen, 2025, Muennighoff et al., 2025, Yeo et al., 2025]. As a result, these models demonstrate increasingly human-like deliberative abilities and can generate extended chains of thought (CoT) to support long-horizon reasoning [Li et al., 2025a, Xu et al., 2025, Chen et al., 2025a].

Despite the remarkable advancements of LLMs' reasoning capabilities, progress has been heavily concentrated in high-resource languages such as English and Chinese [Glazer et al., 2024, Phan

---

\* Equal contribution
† Corresponding author

39th Conference on Neural Information Processing Systems (NeurIPS 2025).

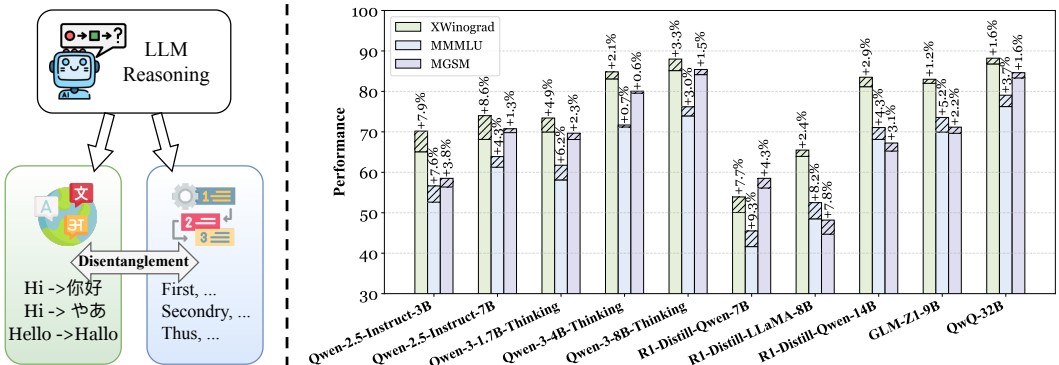

Figure 1: Overview of our hypothesis and findings. **Left:** Motivated by cognitive neuroscience, we hypothesize that reasoning and language processing in LLMs can be disentangled. **Right:** To validate this hypothesis, we perform a causal intervention by removing language-specific components from hidden representations. We evaluate this intervention across 10 open-weight LLMs and 3 multilingual reasoning benchmarks. Results show consistent performance improvements, supporting our claim that disentangling language and reasoning enhances multilingual generalization.

et al., 2025]. While these models exhibit strong performance in these dominant languages, they continue to face challenges in reasoning tasks across low- and mid-resource languages [Karim et al., 2025, Gao et al., 2025]. This growing disparity in multilingual reasoning capabilities raises critical concerns: it hinders the global applicability of LLMs, exacerbates existing linguistic inequities in AI access, and perpetuates the marginalization of underrepresented languages due to limited data and investment [Okolo and Tano, 2024, Ghosh et al., 2025, Qin et al., 2025]. Despite the significance of this issue, the multilingual reasoning gap remains largely under-examined in current research.

In this work, we investigate the interplay between language and reasoning in LLMs, aiming to uncover the key factors that influence cross-lingual reasoning generalization. Our key hypothesis is that *reasoning and language processing can be explicitly disentangled*, allowing reasoning abilities—once acquired in a high-resource language—to transfer more broadly across languages. This idea is supported by cognitive neuroscience findings that the human brain's language network—responsible for comprehension and production—remains largely inactive during reasoning tasks [Monti et al., 2009, 2012, Amalric and Dehaene, 2019], and that human language itself is evolutionarily optimized for communication rather than for reasoning [Fedorenko et al., 2024].

To further validate this hypothesis, we perform causal interventions within LLMs' latent spaces during multilingual reasoning by subtracting language-specific components from each hidden state (§3). This intervention aims to disentangle linguistic features from the underlying reasoning process. Notably, as previewed in Figure 1, we observe consistent performance gains in multilingual reasoning across a diverse set of tasks, including mathematical problem solving [Shi et al., 2023], commonsense inference [Muennighoff et al., 2023], and knowledge-intensive question answering [Hendrycks et al., 2021]. These improvements are robust across all 10 open-weight LLMs, encompassing both reasoning-oriented and general-purpose LLMs. Moreover, the benefits generalize across 11 languages of varying resource availability. This consistent pattern provides compelling empirical evidence that reasoning and linguistic processing can be disentangled, enabling the effective transfer of reasoning capabilities from high-resource to low-resource languages.

To further assess the impact of our intervention, we examine the correlation between the intensity of language-specific components in hidden states and reasoning performance (§4.1). Our analysis reveals that stronger language-specific signals tend to correlate with lower reasoning accuracy, implying that excessive linguistic information may disrupt reasoning processes. We further perform a layer-wise ablation study to pinpoint where language and reasoning are most intertwined (§4.2). We find that language–reasoning decoupling improves performance across almost all layers, but interventions in the upper layers significantly degrade output fidelity, indicating that language-specific signals in later layers are crucial for maintaining language-specific generation. In contrast, low and middle layers provide the best trade-off between reasoning gains and linguistic coherence. To further evaluate the practical value of these findings, we compare our training-free intervention with stan-

dard multilingual post-training methods, including both supervised fine-tuning and reinforcement learning (§4.3). Our training-free intervention-based approach achieves comparable or even superior performance. This suggests that structural disentanglement of language and reasoning can serve as a lightweight and effective alternative, or complement, to post-training, opening up promising directions for future cross-lingual model enhancement.

## 2 Disentangle Language and Reasoning in the Activation Space

In §2.1, we identify language-specific subspaces by isolating components that consistently encode linguistic variation across inputs. In §2.2, we introduce a projection-based intervention that removes these components during inference to disentangle language-specific information from the reasoning process. Finally, in §2.3, we provide empirical evidence that the removed components indeed correspond to language-specific signals, validating the effectiveness of our approach and the plausibility of language–reasoning disentanglement in practice.

### 2.1 Language-Specific Subspace Identification

Assuming the backbone model processes reasoning inputs from $L$ different languages, we compute a mean representation for each language $l$ at every layer as follows:

$$\boldsymbol{m}_l = \frac{1}{n} \sum_{i=1}^{n} \boldsymbol{e}_l^i \tag{1}$$

Here, $\boldsymbol{e}_l^i \in \mathbb{R}^d$ denotes the embedding of the final token from the $i$-th sample in language $l$, and $n$ is the total number of samples for that language. By concatenating the mean vectors $\boldsymbol{m}_l$ for all $L$ languages along the column axis, we construct a mean embedding matrix $\boldsymbol{M} \in \mathbb{R}^{d \times L}$ that characterizes the multilingual latent space.

Building on prior works [Pires et al., 2019, Libovický et al., 2020, Yang et al., 2021], the multilingual latent space $\boldsymbol{M}$ can be decomposed into two orthogonal components: (1) a language-agnostic subspace $\boldsymbol{M}_a$ that captures cross-lingual shared semantics, and (2) a language-specific subspace $\boldsymbol{M}_s$ that encodes language-dependent variations in linguistic expression. Following the formulation of Piratla et al. [2020], Xie et al. [2022], Liu et al. [2025], the decomposition objective is:

$$\min_{\boldsymbol{M}_a, \boldsymbol{M}_s, \boldsymbol{\Gamma}} \quad \left\| \boldsymbol{M} - \boldsymbol{M}_a \mathbb{1}^\top - \boldsymbol{M}_s \boldsymbol{\Gamma}^\top \right\|_F^2$$
$$\text{s.t.} \quad \text{Span}\left(\boldsymbol{M}_a\right) \perp \text{Span}\left(\boldsymbol{M}_s\right), \tag{2}$$

where $\boldsymbol{M}_a \in \mathbb{R}^{d \times 1}$, $\boldsymbol{M}_s \in \mathbb{R}^{d \times r}$, and $\boldsymbol{\Gamma} \in \mathbb{R}^{L \times r}$ represents the language-specific coefficients along the $r$ basis directions of $\boldsymbol{M}_s$. The lower dimensionality of $\boldsymbol{M}_a$ is justified by the observation that shared semantic content across languages is often structurally simpler. In contrast, $\boldsymbol{M}_s$ typically requires higher dimensionality to capture the rich diversity of language-specific features.

The optimal solution to Equation 2 can be efficiently obtained via Singular Value Decomposition (SVD), with the detailed procedure provided in Algorithm 1 in Appendix C. Finally, the identified language-specific subspace $\boldsymbol{M}_s$ serves as the foundation for our subsequent intervention aimed at disentangling language-specific signals from the model's reasoning representations.

### 2.2 Activation Ablation for Language-Reasoning Disentanglement

Given that $\boldsymbol{M}_s$ characterizes the language-specific subspace within the model's activation space, we leverage a ablation mechanism to disentangle language-specific signals from multilingual reasoning. Specifically, for any hidden representation $\boldsymbol{h}$ derived from a multilingual input over a specific reasoning task, we project out its components along the subspace spanned by $\boldsymbol{M}_s$:

$$\hat{\boldsymbol{h}} = \boldsymbol{h} - \lambda \boldsymbol{M}_s^\top \boldsymbol{M}_s \boldsymbol{h} \tag{3}$$

where $\lambda$ is the a coefficient to control the ablation strength. This effectively removes language-specific variations, allowing the remaining representation $\hat{\boldsymbol{h}}$ to better reflect language-agnostic reasoning processes. By default, this projection is applied throughout all the model layers, focusing on the final input token representation.

## 2.3 Verifying the Linguistic Nature of Removed Components

To assess the effectiveness of the projection-based intervention (Equation 3) in disentangling language-specific information from the model's reasoning process, we conduct empirical validation from two complementary perspectives: (1) **representation space visualization**, and (2) **language fidelity** between input and output on a specific reasoning query. We perform this validation on `Qwen-2.5-7B-Instruct`, covering languages at different resource levels: French and Japanese (high-resource), Thai (medium-resource), and Swahili (low-resource).

First, as shown in Figure 2, we visualize the hidden representations of the final token across different languages, before and after applying the language-specific ablation defined in Equation 3. After removing components along the language-specific subspace $M_s$, the language clustering effect is reduced. More notably, representations of non-English languages exhibit a clear tendency to shift toward the English cluster, while English representations remain largely unaffected. This convergence results in tighter cross-lingual clustering and reveals a transition toward more language-invariant representations centered around English. Interestingly, although Qwen is pretrained primarily on both Chinese and English, we observe that even Chinese representations (the green cluster) tend to converge toward English rather than forming their own central anchor. A definitive explanation likely relates to the distributional domi-

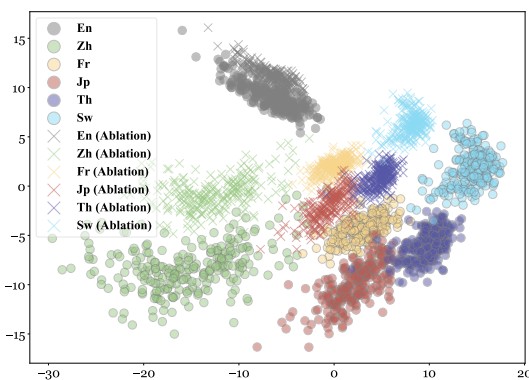

Figure 2: PCA visualization of final-token hidden states before and after projection in Qwen-2.5-7B-Instruct (middle layer). Post-ablation, non-English languages converge toward English, indicating increased language invariance.

nance of English in the pretraining data, though the exact ratios remain unknown. This phenomenon warrants further investigation in future work. Additional visualizations for other model layers and model families are provided in Appendix D.

To further examine how this alignment in representation space affects the model's behavior, we evaluate the *language fidelity* [Holtermann et al., 2024], a metric that quantifies consistency between the input and output languages. We use GlotLID [Kargaran et al., 2023], a multilingual language identifier supporting over 1,600 languages, to detect the language of model-generated responses. As shown in Figure 3, increasing the degree of ablation (i.e., projecting out more components of $M_s$) leads to a substantial drop in language fidelity across all languages: models increasingly default to English in their responses, even when prompted in another language. This behavioral shift mirrors the representational alignment observed in Figure 2, confirming two key findings: (i) the removed components do encode language-specific signals, and (ii) once removed, the model tends to revert to its dominant pretraining language, i.e. English, as the default output anchor, consistent with prior work on multilingual representation aligment [Chen et al., 2023, Wu et al., 2024, Wang et al., 2024, Dumas et al., 2024, Wendler et al., 2024, Chen et al., 2024, Zhao et al., 2024a].

# 3 Causal Intervention within the Activation Space

In this section, we perform causal interventions on the hidden states of LLMs during multilingual reasoning tasks, with the goal of empirically validating our central hypothesis that language and reasoning are functionally separable within LLMs.

**Models** We conduct intervention experiments across a diverse set of LLMs, covering both reasoning and non-reasoning models. Specifically, for non-reasoning (instruction-tuned) models, we include the Qwen-2.5-Instruct family (3B and 7B) [Yang et al., 2024] and the Qwen-3-Instruct series (1.7B, 4B, and 8B) [Team, 2025]. For reasoning-oriented models, we study DeepSeek-R1-Distill (7B and 14B) [Guo et al., 2025], GLM-Z1-9B [GLM et al., 2024], and QwQ-32B [Qwen, 2025]. Notably, for the Qwen-3 series, we employ the `Thinking` mode to enable explicit reasoning.

Table 1: Multilingual reasoning performance on MGSM datasets across different languages, before and after language-reasoning disentanglement within the activation spaces of the backbone models (+ L-R Disentangle). The best results are highlighted in bold. The values in parentheses indicate language fidelity to indicate input-output consistency.

| | High-Resource | | | | | | | Mid-Resource | | Low-Resource | | AVG. |
| | En | Es | Fr | De | Zh | Jp | Ru | Th | Te | Bn | Sw | - |
|---|---|---|---|---|---|---|---|---|---|---|---|---|
| Qwen-2.5-Instruct-3B | 86.0 | 75.2 | 70.8 | 70.0 | 68.8 | 59.6 | 60.8 | 61.6 | 10.0 | 37.2 | 12.4 | 56.36 (90.33%) |
| + L-R Disentangle | 85.6 | **76.8** | **72.0** | **72.0** | **72.4** | **61.2** | **73.6** | **64.8** | **10.8** | **39.6** | **14.8** | **58.51 (91.20%)** |
| Qwen-2.5-Instruct-7B | 92.4 | 82.8 | 78.8 | 77.2 | 82.8 | 72.4 | 81.2 | 79.2 | 36.8 | **67.6** | 16.8 | 69.82 (79.75%) |
| + L-R Disentangle | **92.8** | **84.0** | **79.6** | **80.4** | **84.4** | **73.6** | **82.8** | 79.2 | **37.2** | 64.4 | **20.0** | **70.76 (84.44%)** |
| Qwen-3-1.7B-Thinking | 91.6 | 84.4 | 78.8 | 76.8 | 79.6 | 73.6 | 76.4 | 76.4 | 38.8 | 62.4 | **10.4** | 68.11 (**27.78%**) |
| + L-R Disentangle | 91.6 | **85.2** | **79.2** | **79.2** | **83.2** | **75.2** | **79.2** | **78.8** | **41.2** | **64.8** | 8.8 | **69.67** (27.64%) |
| Qwen-3-4B-Thinking | 95.6 | 88.8 | 79.2 | 83.2 | **88.0** | 81.2 | **85.6** | **85.2** | 72.4 | 68.8 | 32.0 | 79.56 (**27.30%**) |
| + L-R Disentangle | **96.0** | **89.2** | **82.0** | 83.2 | 87.6 | **83.6** | 85.2 | 84.8 | **73.6** | **82.8** | **32.4** | **80.04 (27.31%)** |
| Qwen-3-8B-Thinking | **96.4** | 88.0 | 82.4 | 79.2 | 87.2 | 85.2 | 89.2 | 90.0 | 80.8 | **88.0** | 59.2 | 84.15 (27.24%) |
| + L-R Disentangle | 96.0 | **90.4** | **84.4** | **83.6** | **89.2** | **87.6** | 89.2 | **90.4** | **81.2** | 87.2 | **60.4** | **85.42 (27.27%)** |
| R1-Distill-Qwen-7B | 70.0 | 65.2 | 67.6 | **74.0** | 78.0 | 54.4 | 70.4 | 55.2 | 27.2 | 48.8 | 6.4 | 56.11 (90.98%) |
| + L-R Disentangle | **71.2** | **69.2** | **68.8** | 73.2 | **81.2** | **57.2** | **73.6** | **56.8** | **29.6** | **54.8** | **7.6** | **58.51 (92.18%)** |
| R1-Distill-LLaMA-8B | 79.2 | 51.6 | 51.6 | 49.2 | 70.4 | 52.0 | **56.4** | 38.8 | 12.8 | 26.4 | 3.2 | 44.69 (84.44%) |
| + L-R Disentangle | **84.8** | **56.0** | **55.6** | **52.4** | **73.6** | **55.2** | 56.0 | **46.4** | **14.0** | **29.6** | **6.4** | **48.19 (84.44%)** |
| R1-Distill-Qwen-14B | 70.8 | 71.2 | 74.4 | 75.6 | 85.6 | 68.8 | **82.8** | 76.8 | 20.8 | **67.2** | 23.6 | 65.24 (93.93%) |
| + L-R Disentangle | **71.6** | **73.6** | **75.6** | **76.8** | **86.4** | **73.2** | 82.4 | **81.2** | **26.4** | 66.8 | **25.6** | **67.24 (95.20%)** |
| GLM-Z1-9B | 94.0 | 80.8 | **74.0** | 75.6 | 84.8 | 73.2 | **83.2** | 70.8 | 41.6 | 41.6 | 44.4 | 69.64 (55.96%) |
| + L-R Disentangle | **94.8** | **85.2** | 72.8 | **76.0** | **87.2** | **79.2** | 80.8 | **71.6** | **42.4** | **43.6** | **47.2** | **71.16 (56.04%)** |
| QwQ-32B | 95.6 | 89.2 | 82.8 | 80.4 | 90.0 | 86.4 | 85.6 | **89.2** | 58.4 | 82.4 | **76.4** | 83.31 (56.44%) |
| + L-R Disentangle | **97.6** | **90.8** | **84.4** | **83.6** | **91.2** | **88.0** | 86.4 | 88.8 | **62.0** | **83.6** | 74.8 | **84.62 (60.47%)** |

**Languages**  We select 11 target languages to evaluate the multilingual reasoning under intervention. These languages span diverse linguistic families and resource levels, ensuring broad typological coverage. Specifically, we include high-resource languages such as English (En), Spanish (Es), French (Fr), German (De), Chinese (Zh), Japanese (Jp), and Russian (Ru); medium-resource languages such as Thai (Th) and Telugu (Te); and low-resource languages such as Bengali (Bn) and Swahili (Sw). This selection is designed to balance linguistic diversity with practical considerations of data availability and benchmark coverage.

**Benchmarks**  We evaluate the impact of language-reasoning disentanglement in multilingual reasoning across three major benchmarks, including **MGSM** [Shi et al., 2023]: mathematical reasoning, **XWinograd** [Muennighoff et al., 2023]: commonsense reasoning, and **M-MMLU** [Hendrycks et al., 2021]: knowledge-intensive question answering. We use **Accuracy** as the evaluation metric for all benchmarks. A detailed description of each benchmark is provided in Appendix E.

**Implementation Details**  Our language-reasoning disentanglement are implemented based on the vLLM framework [Kwon et al., 2023] for efficient inference. For each model, we disentangle the language-specific components from mid-layer hidden states and then re-inject them at higher layers. This strategy preserves overall input-output consistency while allowing us to isolate the causal effect of language-specific information on reasoning. A detailed layer-wise analysis of this mechanism is provided in §4.2, and hyperparameter configurations can be found in Appendix F.

**Results and Analysis**  The results for MGSM are presented in Table 1, while the outcomes for XWinograd and M-MMLU are shown in Table 2. We draw the following conclusions:

**Language–reasoning disentanglement consistently improves multilingual reasoning performance.** This improvement holds across all model types and architectures. For non-reasoning models, both the Qwen-2.5 and Qwen-3 Instruct series exhibit clear gains in multilingual reasoning after language–reasoning disentanglement. Similarly, for reasoning models, the intervention proves effective across models trained with different paradigms: both the DeepSeek-R1-Distill series, which are distilled from stronger models via supervised fine-tuning, and models optimized with large-scale reinforcement learning objectives like QwQ-32B, show substantial multilingual improvements.

Table 2: Multilingual reasoning performance on XWinograd and M-MMLU datasets across different languages, before and after language-reasoning disentanglement within the activation spaces of the backbone models (+ L-R Disentangle). The best results are highlighted in bold. The values in parentheses indicate language fidelity to indicate input-output consistency.

| | XWinograd | | | | | | M-MMLU | | | | | | | | |
|---|---|---|---|---|---|---|---|---|---|---|---|---|---|---|---|
| | En | Fr | Zh | Jp | Ru | AVG. | En | Es | Fr | De | Zh | Jp | Bn | Sw | AVG. |
| Qwen-2.5-Instruct-3B | 71.5 | 63.9 | 68.5 | 66.0 | 55.5 | 65.07 (98.31%) | 71.5 | 57.0 | 55.0 | 52.5 | 54.0 | 55.0 | 42.0 | 34.0 | 52.62 (30.56%) |
| + L-R Disentangle | 75.5 | 69.9 | 73.5 | 71.0 | 61.0 | 70.18 (99.52%) | 71.5 | 60.5 | 57.5 | 53.5 | 57.5 | 56.5 | 49.5 | 38.5 | 56.63 (31.06%) |
| Qwen-2.5-Instruct-7B | 73.0 | 62.7 | 73.5 | 78.5 | 53.0 | 68.13 (98.70%) | 74.5 | 69.0 | 70.5 | 65.5 | 66.5 | 62.5 | 52.5 | 29.5 | 61.25 (27.12%) |
| + L-R Disentangle | 78.0 | 73.5 | 79.0 | 82.5 | 57.0 | 74.00 (99.90%) | 78.0 | 74.0 | 70.0 | 66.5 | 68.5 | 65.5 | 56.0 | 35.5 | 63.88 (34.56%) |
| Qwen-3-1.7B-Thinking | 72.5 | 63.9 | 82.0 | 71.5 | 60.0 | 69.97 (60.00%) | 74.5 | 62.5 | 63.5 | 65.0 | 65.5 | 59.5 | 47.0 | 27.5 | 58.12 (12.81%) |
| + L-R Disentangle | 76.0 | 71.1 | 80.5 | 76.0 | 63.5 | 73.42 (59.10%) | 79.0 | 63.5 | 66.0 | 67.0 | 68.5 | 64.5 | 52.5 | 33.0 | 61.75 (13.31%) |
| Qwen-3-4B-Thinking | 83.0 | 81.9 | 84.5 | 87.0 | 79.0 | 83.09 (60.00%) | 81.5 | 76.5 | 75.5 | 77.5 | 74.0 | 75.0 | 67.5 | 42.0 | 71.19 (25.00%) |
| + L-R Disentangle | 86.0 | 83.1 | 88.5 | 88.0 | 78.5 | 84.83 (60.10%) | 81.5 | 77.5 | 78.5 | 76.5 | 74.5 | 73.0 | 70.0 | 40.5 | 71.69 (25.06%) |
| Qwen-3-8B-Thinking | 87.0 | 80.7 | 84.5 | 90.0 | 83.5 | 85.14 (58.80%) | 83.5 | 78.5 | 77.5 | 76.0 | 79.5 | 77.5 | 73.5 | 45.5 | 73.94 (20.69%) |
| + L-R Disentangle | 89.0 | 88.0 | 87.0 | 90.0 | 86.0 | 87.99 (59.70%) | 84.0 | 82.5 | 78.0 | 81.0 | 80.5 | 81.5 | 73.0 | 49.0 | 76.19 (20.13%) |
| R1-Distill-Qwen-7B | 54.5 | 45.8 | 47.0 | 54.0 | 49.0 | 50.06 (40.00%) | 61.0 | 51.5 | 52.0 | 45.5 | 30.0 | 45.5 | 31.0 | 16.5 | 41.63 (24.75%) |
| + L-R Disentangle | 54.5 | 49.4 | 53.0 | 54.5 | 47.5 | 53.90 (50.50%) | 61.0 | 56.5 | 55.0 | 53.0 | 37.5 | 51.5 | 33.0 | 16.5 | 45.50 (24.63%) |
| R1-Distill-LLaMA-8B | 78.0 | 68.7 | 56.5 | 53.5 | 63.0 | 63.93 (75.80%) | 67.5 | 62.5 | 57.5 | 52.5 | 43.0 | 48.5 | 30.0 | 26.5 | 48.50 (19.19%) |
| + L-R Disentangle | 79.5 | 67.5 | 62.5 | 54.5 | 63.5 | 65.49 (76.20%) | 72.0 | 61.0 | 61.0 | 61.0 | 45.5 | 52.5 | 36.0 | 31.0 | 52.50 (19.37%) |
| R1-Distill-Qwen-14B | 90.5 | 78.3 | 74.5 | 85.5 | 77.0 | 81.16 (65.30%) | 81.5 | 78.5 | 75.5 | 77.5 | 64.0 | 76.5 | 60.5 | 31.0 | 68.13 (23.69%) |
| + L-R Disentangle | 91.0 | 79.5 | 81.5 | 87.5 | 78.0 | 83.50 (65.40%) | 85.5 | 80.0 | 80.0 | 80.0 | 66.5 | 78.5 | 62.0 | 36.0 | 71.06 (24.50%) |
| GLM-Z1-9B | 87.5 | 81.9 | 80.5 | 78.0 | 82.0 | 81.99 (41.00%) | 82.5 | 77.5 | 75.0 | 79.0 | 71.5 | 71.0 | 59.0 | 44.0 | 69.94 (24.87%) |
| + L-R Disentangle | 89.0 | 80.7 | 82.5 | 83.0 | 81.0 | 83.00 (40.20%) | 84.5 | 82.0 | 78.5 | 79.0 | 76.0 | 75.5 | 65.5 | 47.5 | 73.56 (25.00%) |
| QwQ-32B | 92.5 | 88.0 | 85.5 | 89.5 | 78.5 | 86.79 (65.30%) | 84.0 | 81.5 | 81.5 | 81.0 | 81.5 | 79.5 | 75.5 | 45.5 | 76.25 (13.19%) |
| + L-R Disentangle | 94.5 | 91.6 | 85.5 | 92.0 | 77.5 | 88.21 (69.90%) | 87.0 | 85.5 | 83.0 | 84.0 | 85.5 | 82.0 | 75.5 | 49.5 | 79.06 (13.25%) |

Moreover, the benefits of language–reasoning disentanglement are even more pronounced on XWinograd and M-MMLU benchmarks that emphasize commonsense inference and knowledge-intensive reasoning. Nearly all models show larger absolute gains in average accuracy after intervention. This suggests that linguistic interference is particularly detrimental in reasoning tasks requiring subtle context integration or factual grounding, and that removing language-specific noise in the representation space can yield stronger improvements under these conditions.

**The performance gains are consistent across languages with different resource levels.** High-resource languages such as English, French, and Chinese show steady improvements, indicating that even well-represented languages benefit from suppressing language-specific interference. More notably, the enhancement is also substantial in medium- and low-resource languages. For example, Swahili—despite being minimally present in pretraining data—achieves accuracy gains exceeding 10% in several models, with some cases more than doubling the original performance. These results demonstrate that our intervention provides balanced benefits across the linguistic spectrum and is also impactful in underrepresented settings, contributing to more equitable multilingual reasoning.

## 4 Deeper Analysis

### 4.1 Language-Specific Activation Negatively Correlates with Reasoning Accuracy

To further understand the impact of language-specific signals on multilingual reasoning, we quantitatively analyze how the intensity of these signals affects model performance. Specifically, we vary the *ablation strength*, defined as the proportion of language-specific components in $M_s$ removed (positive values) or injected back (negative values) during inference. This allows us to observe both the benefits of suppressing and the effects of amplifying language-specific information.

As shown in Figure 3, we report MGSM accuracy alongside two fidelity metrics: reasoning fidelity measures whether the model's intermediate reasoning aligns with the input language, while response fidelity captures whether the final answer is delivered in the same language as the input. There are three representative models involved: Qwen2.5-7B-Instruct, DeepSeek-R1-Distill-7B, and QwQ-32B. We observe a consistent negative correlation between language-specific activation and reasoning performance. As the ablation strength increases (i.e., more language-specific components are removed), MGSM accuracy improves. Conversely, when language-specific components are amplified (negative ablation strength), reasoning performance degrades—most notably in the QwQ model, where accuracy drops steeply as language-specific activation increases.

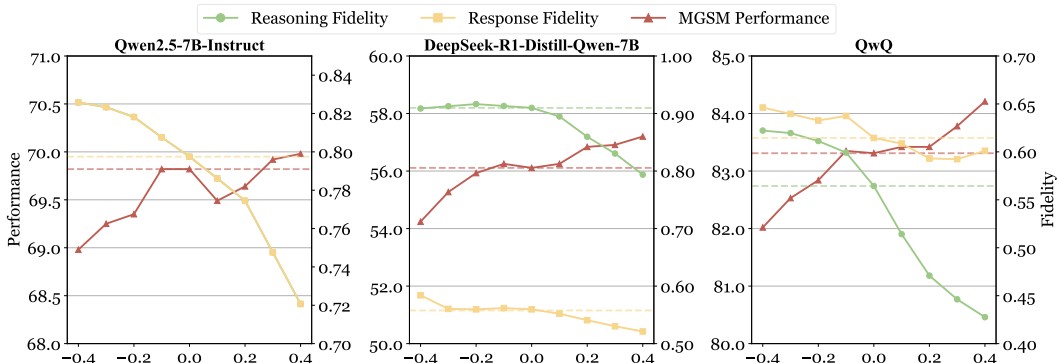

Figure 3: Effects of ablation strength on multilingual reasoning performance and output fidelity. Positive values indicate the proportion of language-specific components in $M_s$ removed, while negative values indicate their injection (reinforcing language-specific signals). We report MGSM accuracy (reasoning performance, red), reasoning fidelity (green), and response fidelity (yellow). Dashed lines indicate the original model's performance without intervention.

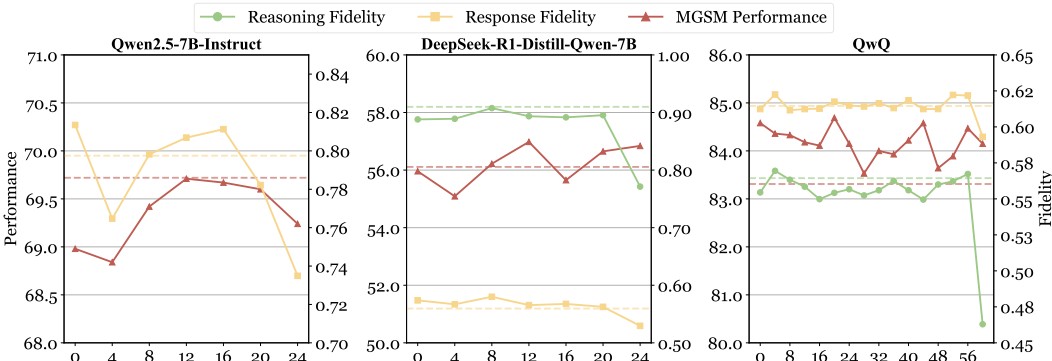

Figure 4: Layer-wise impact of language–reasoning disentanglement on MGSM accuracy and output fidelity. The x-axis denotes the starting layer index of the intervention. Most layers support effective disentanglement that improves reasoning performance. Dashed lines indicate the original model's performance without intervention.

These trends reaffirm our central hypothesis: excessive entanglement of linguistic signals can interfere with multilingual reasoning. By suppressing such components, we guide the model toward more language-invariant internal representations and thereby enhance reasoning performance. At the same time, we observe trade-offs in response fidelity, suggesting that fine-grained control over ablation strength may be required to balance accuracy and output fluency.

We further analyze the effect of ablation strength on each individual language. The reasoning performance trends align with the overall average, improving as language-specific signals are suppressed. However, response fidelity degrades more noticeably in low-resource languages, indicating their higher dependence on language-specific components for fluent generation. Full results and analysis are provided in Appendix G.1.

## 4.2 Layer-wise Effects of Language–Reasoning Disentanglement

To investigate where language and reasoning are most effectively separable, we conduct a layer-wise analysis. Specifically, we apply the disentanglement intervention to different layers of the model, from lower to middle to upper, and evaluate its impact on reasoning performance and output fidelity.

As shown in Figure 4, we observe that ablation applied at most layers, especially from lower to middle depths, consistently improves reasoning accuracy while maintaining stable output fidelity. This suggests that language–reasoning disentanglement is broadly effective across the network and does not require precise targeting to specific layers to yield benefits.

Table 3: Multilingual reasoning accuracy on MGSM across different languages using Qwen-2.5-Instruct-3B, comparing three approaches: baseline, language-reasoning Disentanglement (+ L-R Disentangle), and multilingual post-training (SFT and RL). The best result for each language is highlighted in bold, and the second-best is marked with underline.

| | High-Resource | | | | | | | Mid-Resource | | Low-Resource | | AVG. |
| | En | Es | Fr | De | Zh | Jp | Ru | Th | Te | Bn | Sw | - |
|---|---|---|---|---|---|---|---|---|---|---|---|---|
| Qwen-2.5-Instruct-3B | 86.0 | 74.0 | 70.0 | 70.0 | 69.2 | 56.0 | 71.2 | 61.6 | 8.0 | 35.2 | 12.4 | 55.78 |
| w/ L-R Disentangle | **87.2** | **76.8** | 72.0 | **71.2** | **73.6** | 61.6 | 73.6 | 65.2 | 12.0 | 40.8 | **15.2** | 59.02 |
| w/ SFT | 80.0 | 60.0 | 60.0 | 60.0 | 70.0 | 30.0 | 70.0 | 70.0 | **20.0** | 10.0 | 10.0 | 49.09 |
| w/ RL (PPO) | 82.0 | 72.8 | **75.2** | 70.0 | 72.8 | **63.2** | **74.8** | **70.4** | 14.8 | **53.2** | 14.4 | **60.33** |

However, interventions at the upper layers lead to a sharp drop in both reasoning and response fidelity. This indicates that while the model's core reasoning logic can operate more language-invariantly, the top layers still encode important information for language fluency. Removing language-specific signals at this stage disrupts generation consistency and harms output quality.

These findings reaffirm the central claim that reasoning and language can be disentangled across a wide range of model depths, but highlight that middle layers offer the best trade-off: substantial gains in reasoning performance without sacrificing linguistic coherence. This also aligns with prior findings showing that multilingual models tend to form language-agnostic semantic spaces in their intermediate layers [Wu et al., 2024, Wang et al., 2024].

## 4.3 Comparison with Multilingual Post-training

We compare the performance of models after language-specific ablation with those enhanced via multilingual post-training, including both supervised fine-tuning (SFT) and reinforcement learning (RL) on mathematical reasoning datasets. This comparison serves to evaluate the practical utility of our proposed intervention and further validate the central hypothesis: that disentangling language from reasoning offers an efficient and effective pathway for improving multilingual reasoning—potentially rivaling expensive post-training strategies.

**Experimental Setup**   We use Qwen2.5-3B-Instruct as the base model due to computational constraints; this is the largest model we can post-train on 8×A100 (80GB) GPUs. Both SFT and RL experiments are conducted using the 7,500-sample MATH benchmark [Hendrycks et al., 2021], which provides verifiable ground-truth answers. For SFT, golden reasoning traces are distilled from QwQ-32B, a model optimized for high-quality chain-of-thought reasoning. Since MATH is English-only, we translate both prompts and reasoning traces into the 10 target languages from §3 using the Google Translate API. SFT is implemented via the `LLaMA-Factory` repository [Zheng et al., 2024], while RL training uses PPO [Schulman et al., 2017] implemented in `OpenRLHF` [Hu et al., 2024]. Further training details are provided in Appendix H.

**Results and Analysis**   The comparison between different methods are demonstrated in Table 3. We derive two key insights from this comparison:

**Training-free intervention achieves performance comparable to post-training.**   By disentangling language and reasoning during inference, our training-free intervention not only surpasses supervised fine-tuning, but also achieves results on par with reinforcement learning. This highlights the practical potential of our core finding: disentangling language-specific information from reasoning dynamics can yield substantial multilingual gains—even without additional training.

Moreover, these results point to a promising direction for future work: Combining reasoning–language disentanglement with SFT or RL may offer a principled way to improve multilingual reasoning while mitigating the inefficiencies of current training pipelines.

**RL remains effective in multilingual settings, while SFT yields limited or even negative gains.** Despite its simplicity, PPO-based RL demonstrates strong potential for enhancing multilingual reasoning, achieving consistent gains across both high- and low-resource languages.

In contrast, SFT fails to provide meaningful benefits, and in many cases, significantly degrades performance. We attribute this to two key factors. First, the multilingual mathematical data used for

SFT is obtained via automatic translation, which may introduce inaccuracies and inconsistencies. This aligns with recent findings that mathematical reasoning data is particularly difficult to translate due to its reliance on precise terminology and logical structure [Liu et al., 2024, She et al., 2024, Wang et al., 2025], highlighting a critical data quality bottleneck in multilingual supervision.

Second, recent works have shown that small models often struggle to mimic the step-by-step reasoning behaviors of larger teacher models, even when high-quality traces are available [Li et al., 2025b, Yu et al., 2025a, Zhao et al., 2025a]. Our results corroborate this conclusion to some extent: direct distillation from QwQ-32B into a 3B model fails to reproduce effective reasoning across languages. This underscores the challenges of multilingual distillation and calls for further research into more effective supervision strategies in cross-lingual contexts.

## 5 Related Works

**Multilingual Reasoning of LLMs** Due to the imbalance in language distribution, most LLMs exhibit strong English bias and limited generalization in low- and mid-resource languages [Zhang et al., 2024a, Dou et al., 2025, Zhao et al., 2025b, Zhou et al., 2025].

To address this limitation, prior research has primarily focused on training-time strategies that fall into two broad categories. The first line of work aims to construct high-quality multilingual reasoning datasets [Zhu et al., 2024, She et al., 2024, Wang et al., 2025, Shimabucoro et al., 2025, Ko et al., 2025], while the second leverages supervision signals from high-resource languages (typically English) to enhance reasoning in low-resource settings [She et al., 2024, Zhao et al., 2024b, Huo et al., 2025, Ruan et al., 2025, Fan et al., 2025]. Complementary to these are test-time scaling approaches, which adapt models at inference time without modifying model weights [Qin et al., 2023, Zhang et al., 2024b, Yong et al., 2025, Gao et al., 2025, Tran et al., 2025, Yu et al., 2025b, Son et al., 2025].

In contrast, our work shifts the focus to the internal of multilingual LLMs. By analyzing how LLMs represent language and reasoning in their latent spaces, we uncover structural entanglement that hinders generalization, and propose a lightweight, training-free intervention that improves multilingual reasoning by explicitly disentangling language-specific signals from the reasoning process.

**Mechanical Interpretation of Multilingualism** Recent studies have explored how LLMs process multilingual inputs from two main perspectives: language-specific encoding and language-agnostic abstraction. The former reveals that certain neurons are selectively activated by specific languages, suggesting internal language-specialized subnetworks [Tang et al., 2024, Kojima et al., 2024, Saito et al., 2024, Zhang et al., 2024c]. In contrast, other works show that LLMs form shared semantic spaces across languages, enabling cross-lingual generalization through language-invariant representations [Wu et al., 2024, Wang et al., 2024, Brinkmann et al., 2025, Chen et al., 2025b].

We examine how language-specific signals interact with reasoning processes and find that suppressing such signals improves cross-lingual reasoning. This provides new evidence of their entanglement and highlights the potential of explicit disentanglement as a path to more robust generalization.

**Representation Engineering** Representation-level interventions in LLMs are gaining traction for their transparency and efficiency [Zou et al., 2023]. Grounded in the Linear Representation Hypothesis [Mikolov et al., 2013, Nanda et al., 2023, Park et al., 2024], prior work explores manipulating hidden states at inference time to enhance truthfulness [Li et al., 2023, Campbell et al., 2023, Zhang et al., 2024d] or reduce harmful behavior [Lee et al., 2024, Uppaal et al., 2024, Zhao et al., 2025c]. We adapt this paradigm to multilingual reasoning by targeting language-specific components in the representation space and demonstrate that their removal improves cross-lingual performance—without any additional training or model modification.

## 6 Discussion, Limitations, and Future Work

While our work provides strong empirical evidence for the effectiveness of language–reasoning disentanglement in large language models (LLMs), it also opens up new questions and highlights several important limitations and future directions:

**Why English as the Language-Invariant Anchor?**  In our projection-based intervention, we observe that representations across languages tend to converge toward English—even in bilingual models like Qwen, which are trained with significant Chinese data. This suggests that English serves as the dominant anchor in the model's latent space. While this may reflect the disproportionately high presence of English in pretraining corpora, the exact causes remain opaque due to the lack of transparency in model training data. Future work could benefit from more controlled experiments or synthetic training settings to better understand how anchor languages emerge in multilingual LLMs.

**Connection to Latent Reasoning**  Our findings on language–reasoning disentanglement resonate with the emerging paradigm of latent reasoning in LLMs [Deng et al., 2024, Goyal et al., 2024, Shen et al., 2025a,b]. Recent work by Hao et al. [2024] introduces the Coconut framework, which enables LLMs to perform reasoning within a continuous latent space, rather than relying solely on explicit language tokens. This approach allows models to internally process reasoning steps as high-dimensional representations—termed "continuous thoughts"—before generating any output.

Both our projection-based intervention and the Coconut framework aim to disentangle reasoning processes from language-specific features. While our method removes language-specific components from hidden states to enhance cross-lingual reasoning, Coconut bypasses the need for linguistic expression during intermediate reasoning steps altogether. These complementary approaches suggest that reasoning in LLMs can be more effectively modeled and enhanced by focusing on latent representations, rather than being constrained by surface-level language.

This convergence opens up promising avenues for future research, such as integrating language–reasoning disentanglement techniques with latent reasoning frameworks to further improve the generalization and interpretability of LLMs across diverse tasks and languages.

# 7  Conclusion

In this work, we explore the internal mechanisms underlying multilingual reasoning in large language models. Motivated by cognitive insights, we hypothesize that reasoning and language processing can be explicitly disentangled. Through targeted interventions in the LLMs' activation space, we demonstrate that removing language-specific information significantly improves reasoning performance across languages. Our empirical analysis reveals a strong inverse correlation between language-specific signal strength and reasoning accuracy, and identifies mid-level layers as the most effective location for intervention. Furthermore, we show that our training-free approach achieves results comparable to, and in some cases exceeding, those of multilingual post-training methods. These findings suggest that structural disentanglement offers a lightweight and scalable alternative for enhancing cross-lingual reasoning, and open up promising opportunities to integrate such interventions with future multilingual training and adaptation strategies.

# Acknowledgments

We thank the anonymous reviewers for their comments and suggestions. This work was supported by the New Generation Artificial Intelligence-National Science and Technology Major Project 2023ZD0121100, the National Natural Science Foundation of China (NSFC) via grant 62441614 and 62176078, the Fundamental Research Funds for the Central Universities, and the National Research Foundation, Singapore under its National Large Language Models Funding Initiative (AISG Award No: AISG-NMLP-2024-002). Any opinions, findings and conclusions or recommendations expressed in this material are those of the author(s) and do not reflect the views of National Research Foundation, Singapore.

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

# A Further Discussion, Limitations, and Future Work

While our work provides strong empirical evidence for the effectiveness of language–reasoning disentanglement in large language models (LLMs), it also opens up new questions and highlights several important limitations and future directions:

**Trade-off Between Reasoning and Language Fidelity.** Although language-specific ablation improves reasoning accuracy, we also observe a degradation in language fidelity, especially when intervention is applied in higher layers. This trade-off highlights the challenge of balancing language-invariance with surface-level coherence. Future efforts could explore more fine-grained or dynamic projection mechanisms that adaptively control which components are suppressed and which are preserved based on task goals or input context.

**Applicability Beyond Reasoning.** While our focus is on multilingual reasoning tasks, the principle of disentangling task-relevant and language-specific signals may extend to other high-level cognitive abilities in LLMs, such as planning, reflection, or explanation generation. Investigating how this approach generalizes to other domains, and whether other types of entangled information can be removed, offers a promising avenue for broader interpretability-guided model improvement.

**From Post-hoc Intervention to Training-time Integration.** Our method operates at inference time and requires no model modification, which makes it lightweight and broadly applicable. However, the benefits of projection-based disentanglement could potentially be amplified if integrated into training objectives. For example, encouraging intermediate representations to be language-invariant during supervised fine-tuning or reinforcement learning could improve generalization more systematically. Future work could explore hybrid objectives that explicitly regularize the disentanglement of reasoning and language components during model optimization.

**Toward Principled Multilingual Model Diagnosis.** Finally, our findings suggest that analyzing language-specific activation patterns offers a diagnostic signal for cross-lingual model behavior. This could form the basis for more principled evaluation frameworks that go beyond surface-level accuracy, helping to reveal when and where models fail to generalize across languages.

# B Broader Impact

As LLMs are increasingly deployed in global applications, ranging from education to healthcare and decision support, ensuring their equitable performance across languages has become both a technical and ethical imperative. However, most recent progress in reasoning-enhanced LLMs remains concentrated in English and a few high-resource languages. This reinforces existing disparities in AI access, limiting the utility of advanced models for speakers of low- and mid-resource languages.

Our work addresses this issue by investigating the internal mechanisms behind multilingual reasoning in LLMs and introducing a lightweight, training-free approach to enhance reasoning capabilities across languages. By explicitly performing language-reasoning disentanglement, we show that reasoning can be made more language-invariant—without requiring costly additional data or retraining. This has the potential to democratize access to strong reasoning models for underrepresented languages, particularly in settings where resources for large-scale finetuning are limited.

In addition, our analysis contributes to the broader goal of making LLMs more interpretable and controllable. Rather than relying solely on post-hoc evaluation, we explore how internal representations can be causally intervened upon to improve specific capabilities. This aligns with the growing movement toward transparent, mechanism-driven AI development and offers tools that can be integrated into responsible deployment pipelines.

Nonetheless, our intervention raises new questions about fairness and control: the choice to remove language-specific features must be carefully balanced against preserving cultural and linguistic identity. Future work should examine the societal implications of language–reasoning disentanglement in real-world multilingual applications, and how such techniques may interact with biases in training data or user experience.

---
**Algorithm 1:** Language Subspace Probing
---
**In:** languages' mean embeddings $M$, rank of subspace $r$
**Out:** language-agnostic subspace $M_a$, language-specific subspace $M_s$, coordinates $\Gamma$

```
/* 1) Approximate M in low rank                                    */
```
1   $M'_a \leftarrow \frac{1}{d} M \mathbb{1}$;
2   $M'_s, \_, \Gamma' \leftarrow \text{Top-}r \text{ SVD}\left(M - M'_a \mathbb{1}^\top\right)$;
3   $M' \leftarrow M'_a \mathbb{1}^\top + M'_s {\Gamma'}^\top$;
```
/* 2) Force orthogonality                                          */
```
4   $M_a \leftarrow \frac{1}{\|{M'}^+ \mathbb{1}\|^2} {M'}^+ \mathbb{1}$;
5   $M_s, \_, \Gamma \leftarrow \text{Top-}r \text{ SVD}\left(M' - M_a \mathbb{1}^\top\right)$
---

Overall, we hope that this work inspires more inclusive and principled approaches to multilingual reasoning, helping to bridge the capability gap across languages and contributing to the development of more equitable, transparent, and linguistically fair AI systems.

## C   Probing for Language Subspace

The optimal solution of Equation 2 can be computed efficiently via Singular Value Decomposition (SVD). Algorithm 1 presents the detailed procedure. Readers interested in more details can consult the proof provided in Xie et al. [2022]. The only hyperparameter $r < L$ controls the amount of language-specific information captured by the identified subspace. The larger $r$ is, the more language-specific signals we can identify.

## D   Representation Space Visualization

To complement the main analysis presented in Section 2.3, we provide additional visualizations of final-token hidden representations across multiple models and layers, before and after applying the projection-based ablation defined in Equation 3. These results offer further evidence of the role played by language-specific subspaces in shaping multilingual representations.

**Layer-wise Comparison in Qwen-2.5-7B-Instruct.**   Figure 5 shows the visualizations of Qwen-2.5-7B-Instruct at three different layer depths: lower, middle, and upper. Across all layers, we observe a consistent trend—after projection, multilingual representations become less language-specific and tend to converge toward the English cluster. However, the degree of convergence varies: the middle layers exhibit the most prominent language alignment, while the lower layers show weaker separation to begin with, and the upper layers retain stronger language-specific traits even after ablation. These results suggest that while the disentanglement effect holds across the model, the middle layers offer the most effective separation between language and reasoning representations.

**Visualizations in Qwen-2.5-3B-Instruct, Qwen-3-8B-Thinking, R1-Distill-Qwen-7B and QwQ-32B.**   Figures 6, Figures 7, Figures 8 and 9 present analogous visualizations for R1-Distill-Qwen-7B and QwQ-32B, respectively. The same convergence trend can be observed: representations of various languages tend to collapse toward the English representation after projection, especially in the middle layers. This confirms that the observed phenomenon generalizes beyond a single model family or training paradigm.

## E   Benchmark

We evaluate the impact of removing language-specific information on multilingual reasoning across three major benchmarks, including:

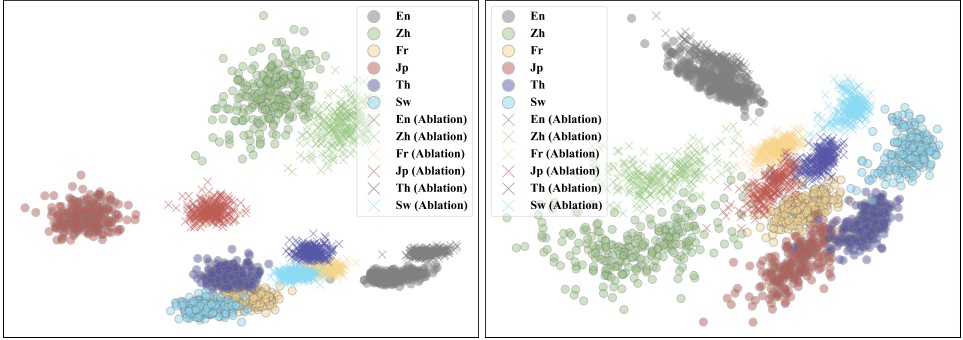

(a) PCA visualization of final-token hidden states before and after ablation in Qwen-2.5-7B-Instruct (5, low layer).

(b) PCA visualization of final-token hidden states before and after ablation in Qwen-2.5-7B-Instruct (14, middle layer).

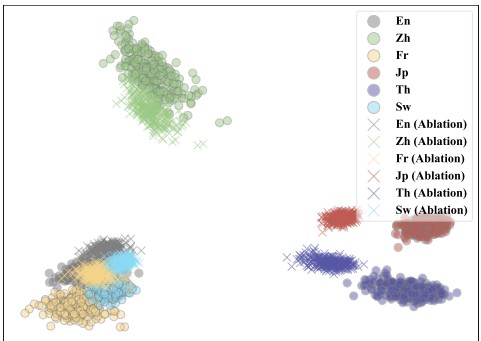

(c) PCA visualization of final-token hidden states before and after ablation in Qwen-2.5-7B-Instruct (27, top layer).

Figure 5: Layer-wise PCA visualizations in Qwen-2.5-7B-Instruct. Each subfigure shows hidden states at lower, middle, and upper layers, before and after projection.

- **MGSM** [Shi et al., 2023]:[3] The Multilingual Grade School Math Benchmark (MGSM) is a dataset designed to evaluate the reasoning abilities of large language models in multilingual settings. It comprises 250 grade-school math problems, originally from the GSM8K dataset, each translated by human annotators into 10 diverse languages: English, Spanish, French, German, Russian, Chinese, Japanese, Thai, Swahili, Bengali, and Telugu . These problems require multi-step reasoning, making MGSM a valuable resource for assessing models' capabilities in mathematical problem-solving across different languages. The dataset includes both the questions and their step-by-step solutions, facilitating comprehensive evaluation of multilingual reasoning performance.

- **XWinograd** [Muennighoff et al., 2023]:[4] A well-established tool for evaluating coreference resolution (CoR) and commonsense reasoning (CSR) capabilities of computational models. The dataset is the translation of the English Winograd Schema datasets and it adds 488 Chinese schemas from CLUEWSC2020 [Xu et al., 2020], totaling 6 languages. Formulated as a fill-in-a-blank task with binary options, the goal is to choose the right option for a given sentence which requires commonsense reasoning. In our experimental setup, this benchmark covers English (En), French (Fr), Chinese (Zh) and Japanese (Jp) and Russian (Ru).

- **M-MMLU** [Hendrycks et al., 2021, Lai et al., 2023]:[5] A benchmark designed to measure knowledge acquired during pretraining by evaluating models exclusively in zero-shot and few-shot settings. The datasets is a machine translated version of the MMLU dataset by GPT-3.5-turbo and covers 34 languages. This is a massive multitask test consisting of multiple-choice questions

---

[3] https://huggingface.co/datasets/juletxara/mgsm
[4] https://huggingface.co/datasets/Muennighoff/xwinograd
[5] https://huggingface.co/datasets/alexandrainst/m_mmlu

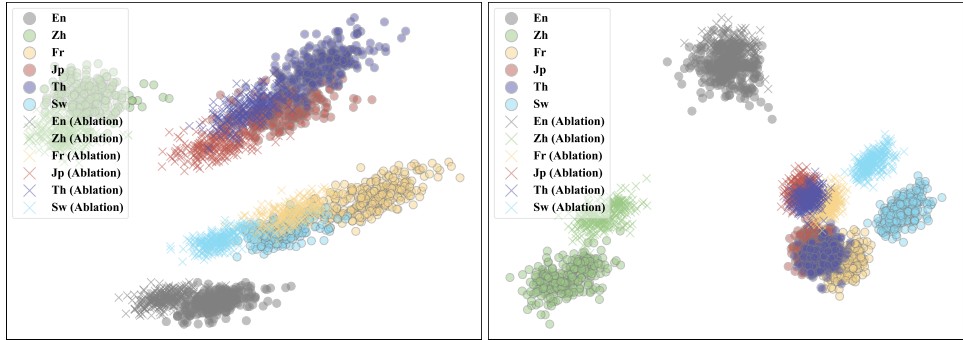

(a) PCA visualization of final-token hidden states before and after ablation in Qwen-2.5-3B-Instruct (2, low layer).

(b) PCA visualization of final-token hidden states before and after ablation in Qwen-2.5-3B-Instruct (16, middle layer).

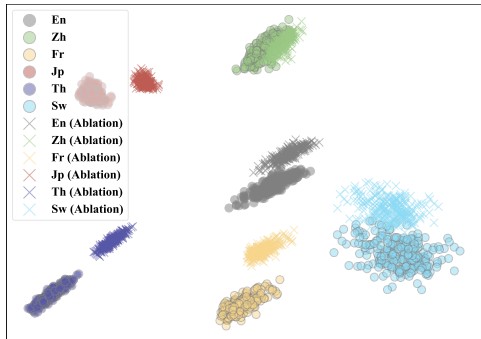

(c) PCA visualization of final-token hidden states before and after ablation in Qwen-2.5-3B-Instruct (35, top layer).

Figure 6: Layer-wise PCA visualizations in Qwen-2.5-3B-Instruct. Each subfigure shows hidden states at lower, middle, and upper layers, before and after projection.

from various branches of knowledge. To attain high accuracy on this test, models must possess extensive world knowledge and problem solving ability. In our experimental setup, this benchmark covers English (En), Spanish (Es), French (Fr), German (De), Chinese (Zh), Japanese (Jp), Bengali (Bn) and Swahili (Sw).

# F    Implementation Details of Causal Intervention

We use 7,500 math problems from the Google Translate version of the MATH dataset [Hendrycks et al., 2021], translated into 10 languages—Bengali (bn), German (de), Spanish (es), French (fr), Japanese (jp), Russian (ru), Swahili (sw), Telugu (te), Thai (th), and Chinese (zh)—as the data for extracting the language-specific subspace. For each input, we ablate activations only on the tokens in the prompt. In the main experiments, we perform a grid search to find the best results, applying a $\lambda$ in the range of 0 to 0.4 at the middle layers and -0.4 to 0 at the higher layers. For various LLMs, the specific range of middle and high layers we have selected are shown in Table 4. Furthermore, in the analysis of ablation strength presented in Section 4.1, our focus is on evaluating model performance in the middle layers and examining language fidelity in the higher layers. Also, We conduct significance tests on the multilingual results, showing that our method yields statistically significant gains ($p < 0.05$) on most benchmarks. We believe this further supports the effectiveness of our approach, especially given its low cost and wide language coverage.

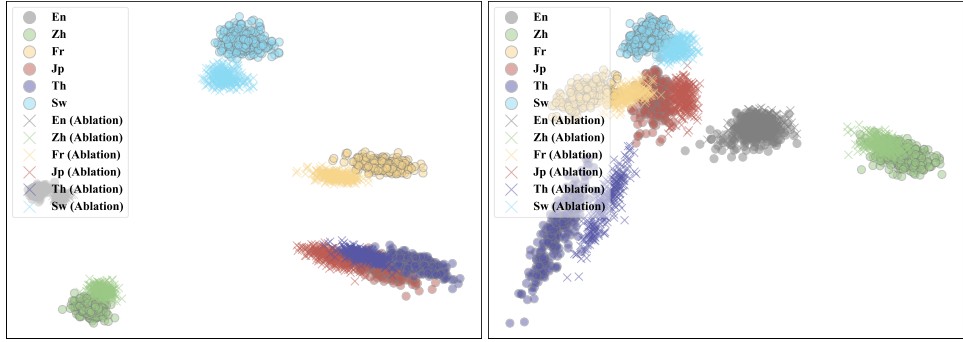

(a) PCA visualization of final-token hidden states before and after ablation in Qwen-3-8B-Thinking (5, low layer).

(b) PCA visualization of final-token hidden states before and after ablation in Qwen-3-8B-Thinking (17, middle layer).

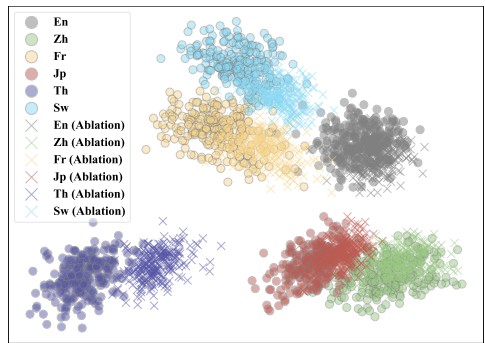

(c) PCA visualization of final-token hidden states before and after ablation in Qwen-3-8B-Thinking (35, top layer).

Figure 7: Layer-wise PCA visualizations in Qwen-3-8B-Thinking. Each subfigure shows hidden states at lower, middle, and upper layers, before and after projection.

Table 4: The exact layer ranges for each model, detailing the total number of layers along with specified middle and higher layer ranges.

| Model Name | Number of Layers | Middle Layers | Higher Layers |
|---|---|---|---|
| Qwen-2.5-Instruct-3B | 36 | 12-26 | 27-35 |
| Qwen-2.5-Instruct-7B | 28 | 10-19 | 20-27 |
| Qwen-3-1.7B-Thinking | 28 | 10-19 | 20-27 |
| Qwen-3-4B-Thinking | 36 | 12-26 | 27-35 |
| Qwen-3-8B-Thinking | 36 | 12-26 | 27-35 |
| R1-Distill-Qwen-7B | 28 | 10-19 | 20-27 |
| R1-Distill-LLama-8B | 32 | 12-22 | 23-31 |
| R1-Distill-Qwen-14B | 48 | 16-33 | 34-47 |
| GLM-Z1-9B | 40 | 12-30 | 31-39 |
| QwQ-32B | 64 | 20-46 | 47-63 |

# G  Additional Experimental Results

## G.1  Correlation Analysis on Each Language

Figure 10 presents the effects of varying ablation strength on both reasoning performance and output fidelity for each of the 10 evaluation languages, using QwQ-32B as the backbone model. Overall, we observe that increasing ablation strength—i.e., removing more language-specific components—leads to improved reasoning performance across most languages, consistent with the ag-

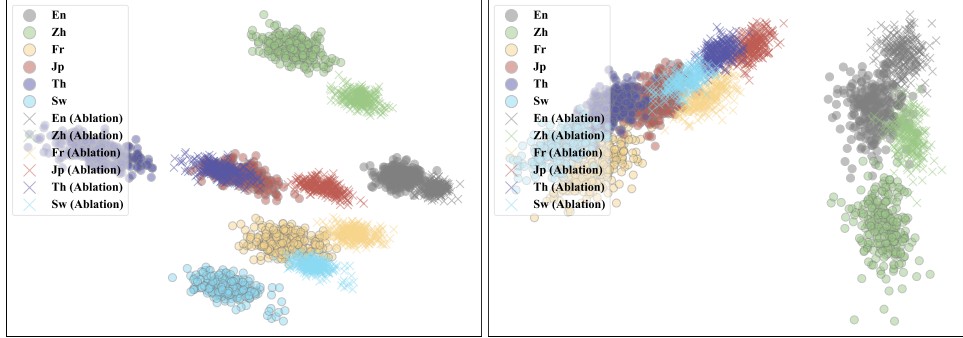

(a) PCA visualization of final-token hidden states before and after ablation in R1-Distill-Qwen-7B (5, low layer).

(b) PCA visualization of final-token hidden states before and after ablation in R1-Distill-Qwen-7B (14, middle layer).

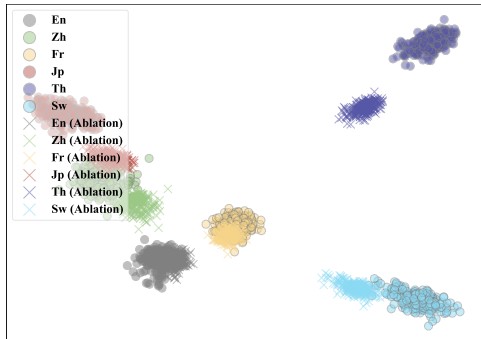

(c) PCA visualization of final-token hidden states before and after ablation in R1-Distill-Qwen-7B (27, top layer).

Figure 8: Layer-wise PCA visualizations in R1-Distill-Qwen-7B. Each subfigure shows hidden states at lower, middle, and upper layers, before and after projection.

gregate trend discussed in Section 4.1. However, the fidelity curves show more nuanced patterns, particularly across languages of different resource levels.

**High-resource languages (En, Es, Fr, Zh, Jp, Ru).** In high-resource languages, reasoning performance is relatively stable or improves steadily with increased ablation. Output fidelity remains high and shows only minor degradation (e.g., English and Chinese maintain near-perfect fidelity throughout). This indicates that high-resource languages are less dependent on language-specific activations for surface realization, likely due to stronger coverage in pretraining.

**Mid- and low-resource languages (Th, Bn, Sw).** In contrast, mid- and low-resource languages exhibit sharper trade-offs. For Thai, Bengali, and Swahili, fidelity drops steeply as ablation strength increases—despite clear gains in reasoning accuracy. For instance, in Thai, output fidelity declines from 0.9 to below 0.4 at high ablation levels, even as performance improves by over 10 points. This suggests that these languages rely more heavily on language-specific signals to maintain fluent generation, possibly due to weaker anchoring in the shared representation space.

**Implication.** These findings highlight the need for a balanced or adaptive ablation strategy in multilingual settings: while removing language-specific components can enhance reasoning performance, overly aggressive suppression may compromise output fluency, especially for underrepresented languages. Future work could explore language-aware projection schedules or hybrid control mechanisms to dynamically balance reasoning abstraction and language preservation.

We further replicate this per-language analysis on two additional models: R1-Distill-Qwen-7B and Qwen-2.5-Instruct-7B, as shown in Figure 11 and

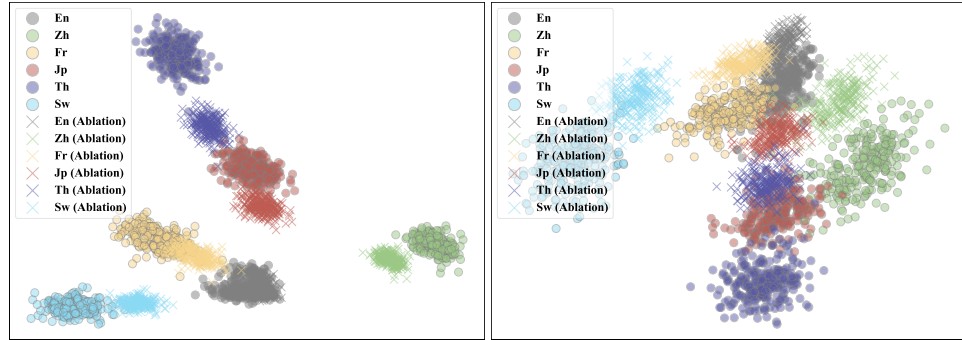

(a) PCA visualization of final-token hidden states before and after ablation in QwQ-32B (12, low layer).

(b) PCA visualization of final-token hidden states before and after ablation in QwQ-32B (35, middle layer).

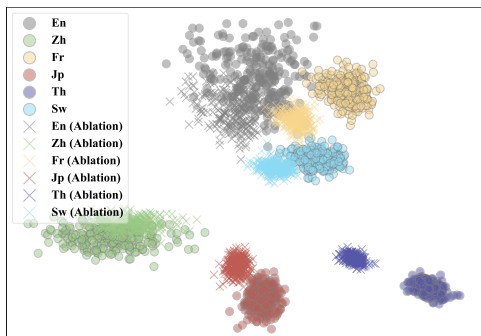

(c) PCA visualization of final-token hidden states before and after ablation in QwQ-32B (63, top layer).

Figure 9: Layer-wise PCA visualizations in QwQ-32B. Each subfigure shows hidden states at lower, middle, and upper layers, before and after projection.

Figure 12, respectively. Across both models, we observe consistent trends with those reported for QwQ-32B: increasing ablation strength generally improves multilingual reasoning performance, while response fidelity declines more significantly in lower-resource languages.

These results reinforce the generality of our findings across different models and training paradigms. They further suggest that the entanglement between language-specific activation and reasoning is a robust phenomenon, and that language–reasoning disentanglement can benefit diverse LLMs—though it may require language-aware tuning to preserve fluency in less represented languages.

### G.2 Layer-wise Analysis on More Models

To further verify the robustness of our findings, we extend the layer-wise analysis introduced in Section 4.2 to additional models, including `Qwen-2.5-3B-Instruct`, `Qwen-3-1.7B`, `Qwen-3-8B`, and `R1-Distill-Qwen-14B`. For each model, we apply the same intervention procedure by ablating language-specific components at different layer depths—specifically lower, middle, and upper layers—and evaluate the effects on multilingual reasoning performance and output fidelity. Results are shown in Figure 13.

The results consistently align with earlier observations: (1) **Performance improves across all models when language-specific signals are removed at middle layers**, reinforcing the hypothesis that reasoning representations are most disentangled from language at this depth; (2) **Upper-layer ablation often degrades output fidelity**, particularly in high-resource languages, indicating that language-specific features in later layers are critical for surface realization.

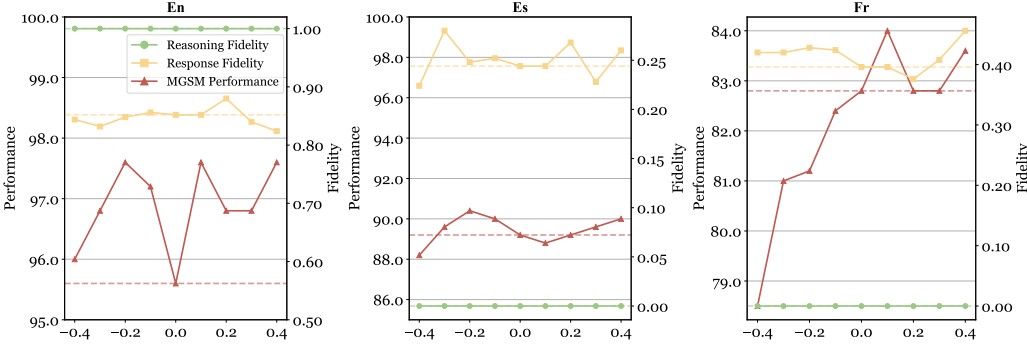

(a) Effects of ablation strength on English reasoning performance and output fidelity.

(b) Effects of ablation strength on Spanish reasoning performance and output fidelity.

(c) Effects of ablation strength on French reasoning performance and output fidelity.

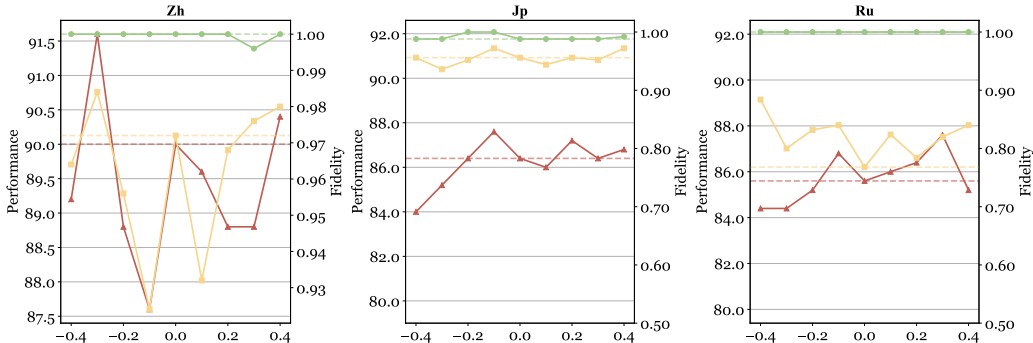

(d) Effects of ablation strength on Chinese reasoning performance and output fidelity.

(e) Effects of ablation strength on Japanese reasoning performance and output fidelity.

(f) Effects of ablation strength on Russian reasoning performance and output fidelity.

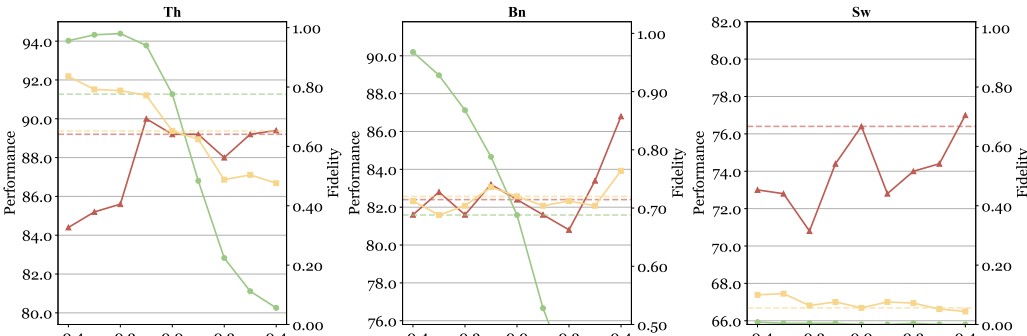

(g) Effects of ablation strength on Thailand reasoning performance and output fidelity.

(h) Effects of ablation strength on Bengali reasoning performance and output fidelity.

(i) Effects of ablation strength on Swahili reasoning performance and output fidelity.

Figure 10: Effects of ablation strength on each language. The backbone is QwQ-32B.

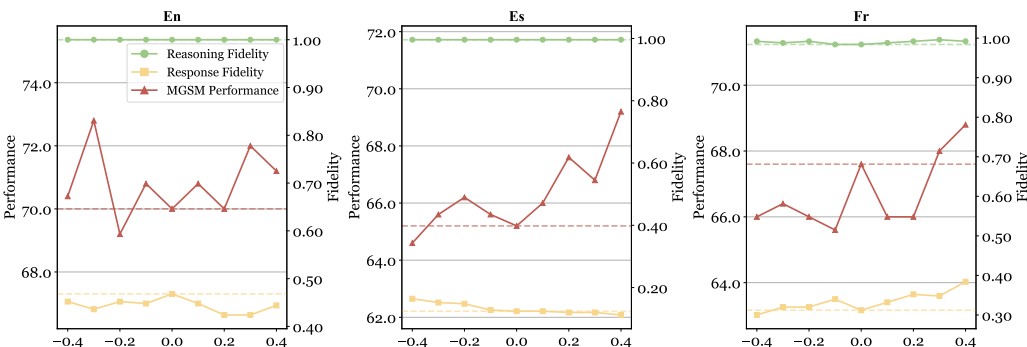

(a) Effects of ablation strength on English reasoning performance and output fidelity.

(b) Effects of ablation strength on Spanish reasoning performance and output fidelity.

(c) Effects of ablation strength on French reasoning performance and output fidelity.

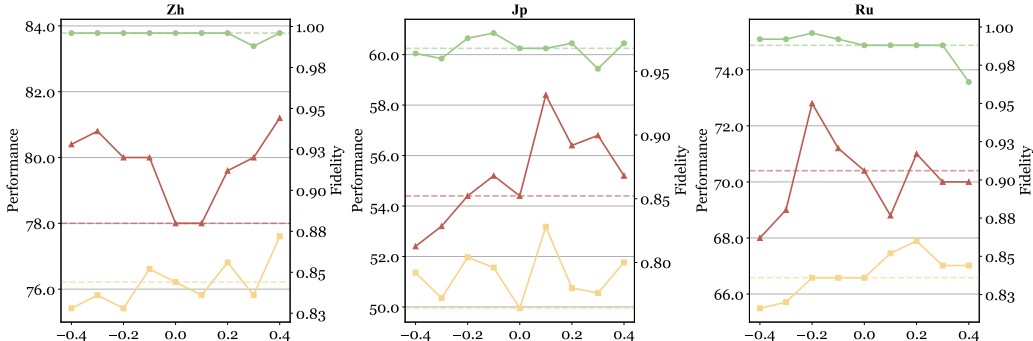

(d) Effects of ablation strength on Chinese reasoning performance and output fidelity.

(e) Effects of ablation strength on Japanese reasoning performance and output fidelity.

(f) Effects of ablation strength on Russian reasoning performance and output fidelity.

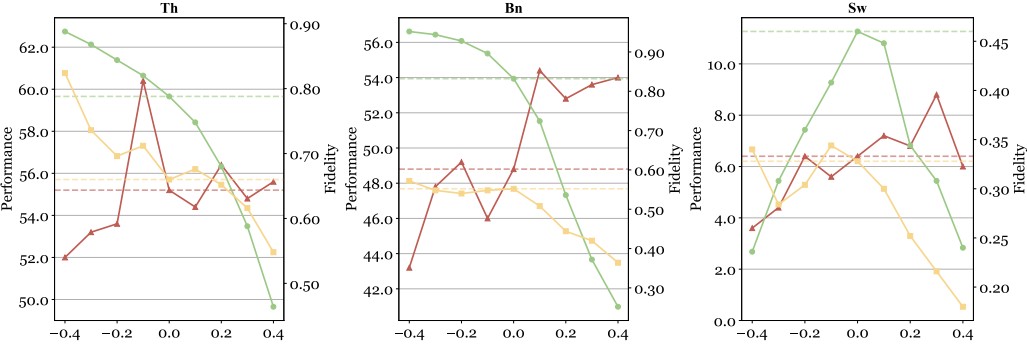

(g) Effects of ablation strength on Thailand reasoning performance and output fidelity.

(h) Effects of ablation strength on Bengali reasoning performance and output fidelity.

(i) Effects of ablation strength on Swahili reasoning performance and output fidelity.

Figure 11: Effects of ablation strength on each language. The backbone is R1-Distill-Qwen-7B.

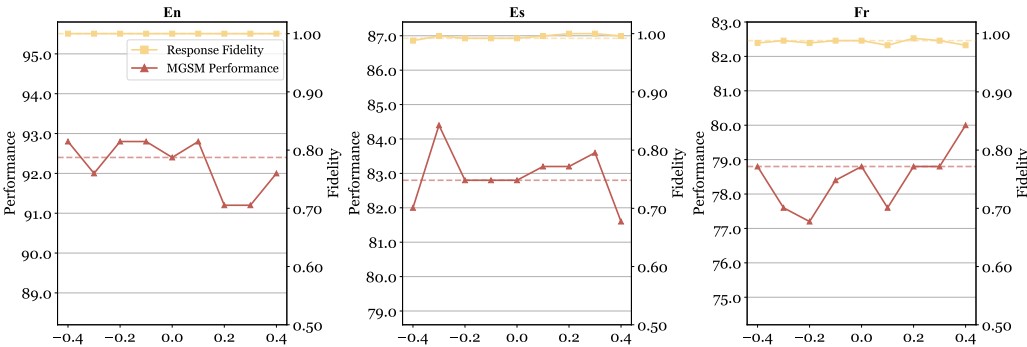

(a) Effects of ablation strength on English reasoning performance and output fidelity.

(b) Effects of ablation strength on Spanish reasoning performance and output fidelity.

(c) Effects of ablation strength on French reasoning performance and output fidelity.

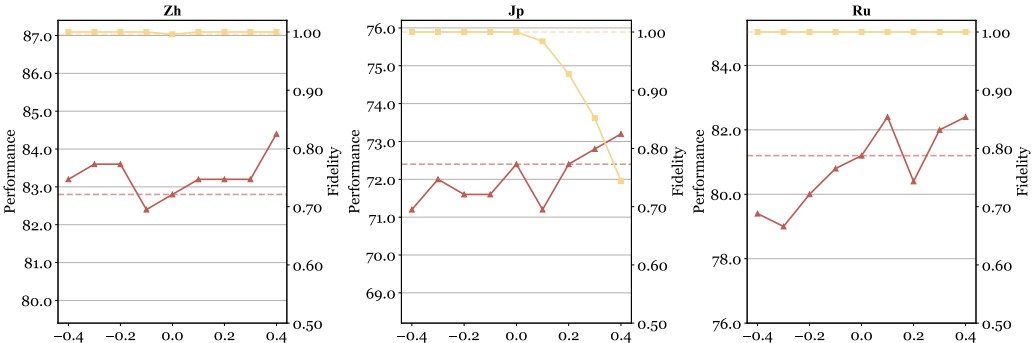

(d) Effects of ablation strength on Chinese reasoning performance and output fidelity.

(e) Effects of ablation strength on Japanese reasoning performance and output fidelity.

(f) Effects of ablation strength on Russian reasoning performance and output fidelity.

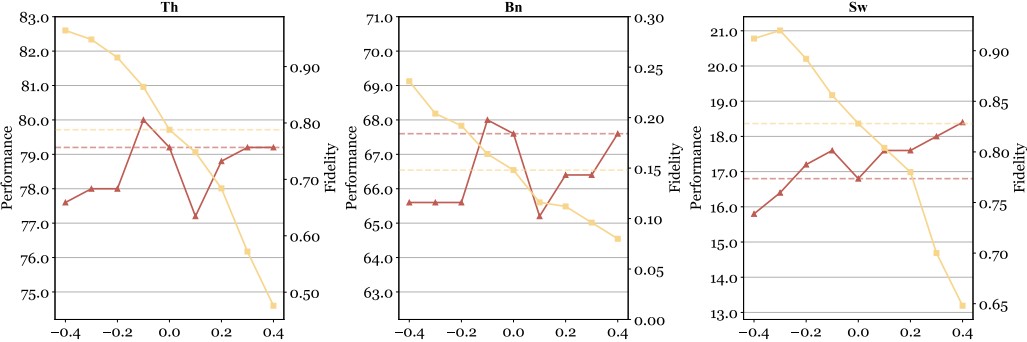

(g) Effects of ablation strength on Thailand reasoning performance and output fidelity.

(h) Effects of ablation strength on Bengali reasoning performance and output fidelity.

(i) Effects of ablation strength on Swahili reasoning performance and output fidelity.

Figure 12: Effects of ablation strength on each language. The backbone is Qwen-2.5-7B-Instruct.

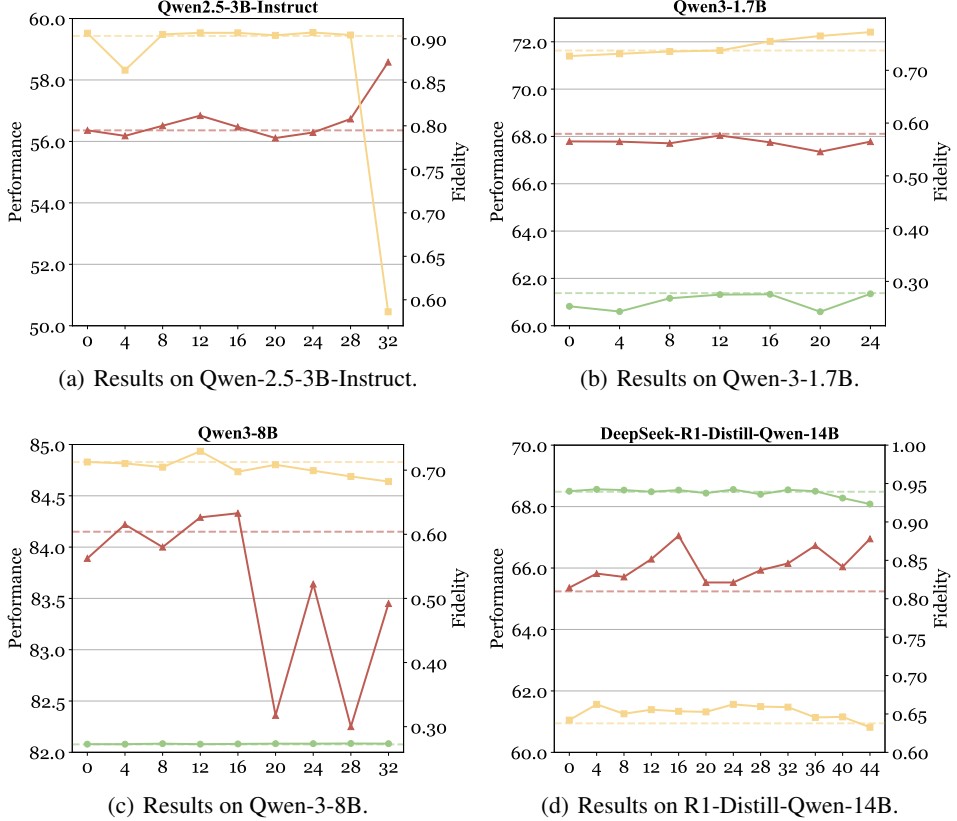

(a) Results on Qwen-2.5-3B-Instruct.

(b) Results on Qwen-3-1.7B.

(c) Results on Qwen-3-8B.

(d) Results on R1-Distill-Qwen-14B.

Figure 13: Layer-wise impact of language–reasoning disentanglement on MGSM accuracy and output fidelity. The x-axis denotes the starting layer index of the intervention. Most layers support effective disentanglement that improves reasoning performance.

These consistent trends across model families and sizes further support the generality of our conclusions and suggest that middle-layer intervention is a broadly effective strategy for improving multilingual reasoning in LLMs.

### G.3 Impact of Dataset Difficulty and Translation Quality

While prior experiments primarily focus on established multilingual reasoning datasets such as MGSM, these benchmarks may not fully capture the difficulty spectrum of modern reasoning tasks. To more comprehensively assess our approach under varying task complexity and translation fidelity, we further evaluate on **PolyMath** [Wang et al., 2025], a recently introduced human-verified multilingual dataset for mathematical reasoning.

The **PolyMath** dataset contains problems annotated with three difficulty levels—low, medium, and high—allowing us to examine reasoning robustness across linguistic and cognitive complexity. For each target language, we randomly sample 50 questions per difficulty level and compute a difficulty-weighted accuracy score. Experiments are conducted on two representative models, `DeepSeek-R1-Distill-Qwen-7B` and `Qwen-2.5-Instruct-7B`, both before and after language–reasoning disentanglement.

Results in Table 5 show that higher-quality and difficulty-balanced datasets substantially reduce the reasoning performance gap between English and other high-resource languages (e.g., Spanish, French). However, mid- and low-resource languages such as Swahili and Telugu continue to underperform, suggesting that structural capability disparities persist even with improved data. Importantly, our intervention helps close this gap: after language–reasoning disentanglement, weaker languages (e.g., Russian in DeepSeek, Thai in Qwen) exhibit notable gains, achieving performance

Table 5: Multilingual reasoning performance on PolyMath datasets across different languages, before and after language-reasoning disentanglement within the activation spaces of the backbone models (+ L-R Disentangle). The best results are highlighted in bold. The values in parentheses indicate language fidelity to indicate input-output consistency.

| | High-Resource | | | | | | | Mid-Resource | | Low-Resource | | AVG. |
| | En | Es | Fr | De | Zh | Jp | Ru | Th | Te | Bn | Sw | - |
|---|---|---|---|---|---|---|---|---|---|---|---|---|
| Qwen-2.5-Instruct-7B | 23.14 | 23.14 | 22.29 | **21.43** | 18.29 | 20.00 | **28.86** | 16.57 | 14.29 | 17.14 | 9.43 | 19.51 (42.00%) |
| + L-R Disentangle | **25.71** | **28.86** | **25.14** | 20.57 | **22.00** | **23.43** | 28.00 | **21.43** | **17.14** | **18.57** | **16.29** | **22.47** (**42.00**%) |
| R1-Distill-Qwen-7B | 48.86 | 46.29 | 41.14 | 49.14 | 45.14 | **45.71** | 39.71 | **42.29** | 28.29 | 42.29 | 21.14 | 40.91 (41.94%) |
| + L-R Disentangle | **50.29** | **50.86** | **48.00** | **51.14** | **46.57** | 43.43 | **46.86** | 41.71 | **34.00** | **45.14** | **21.71** | **43.31** (**44.18**%) |

levels comparable to stronger languages. These findings demonstrate both the practical value and the robustness of our approach under more controlled and linguistically diverse evaluation conditions.

# H  Configuration for Post-training

**Supervised Fine-tuning (SFT)**   All training experiments are conducted on eight A100 GPUs using the LLaMA-Factory repository [Zheng et al., 2024]. For distributed training, we leverage the DeepSpeed [Rasley et al., 2020] framework with ZeRo-2 optimization. The optimizer is AdamW. We train the model for 3 epochs on the multilingual version of the MATH dataset (7,500 samples), with a total batch size of 32, a learning rate of 1e-5 and the max context length of 8,192. The learning rate follows a cosine annealing schedule with 10% warm-up steps.

**Reinforcement Learning (RL)**   For RL, we apply the PPO algorithm [Schulman et al., 2017] using the `OpenRLHF` framework [Hu et al., 2024]. Each training run consists of 3 episodes, with 4 rollouts per sample. We train for 1 epoch on the same multilingual MATH dataset, using a batch size of 96. The maximum input length is set to 1,024 tokens, and the maximum output length is 8,192 tokens to accommodate long reasoning traces. The actor model is optimized with a learning rate of 5e-7, while the critic uses a higher learning rate of 9e-6. We apply a KL penalty coefficient of 0.01 to stabilize training and prevent the actor from drifting too far from the initial policy.

