# OpenReview forum: "When Less Language is More: Language-Reasoning Disentanglement Makes LLMs Better Multilingual Reasoners"
_NeurIPS.cc/2025/Conference — NeurIPS 2025 spotlight_

### Official Review · Reviewer_EnSg · 2025-06-25

**Clarity:** 4
**Significance:** 2
**Originality:** 3
**Rating:** 4
**Confidence:** 3

**Summary:**

This paper studies multilingual reasoning in LLMs. The issue in multilingual reasoning is that the reasoning ability is usually better in high-resource languages such as English and Chinese.

The paper argues that (i) language and (ii) reasoning are separate components in LLMs (similar to in humans as studied in neuroscience), and thus they can be disentangled to enhance multilingual reasoning.

Specifically, the paper takes the embedding of the final token as a representation of a document (in a certain language), and takes the average over all documents from the same language to obtain the mean embedding vector $\textbf{m}_l$. With the mean vectors of all languages, SVD is applied to decompose them into two orthogonal spaces: (1) $M_a$ shared semantic content (of lower rank = 1) and (2) $M_s$ language-specific. To remove language-specific features, any hidden representation $h$ can be projected to the $M_s$ subspace, and then subtracted from $h$ to obtain the language-agnostic part. The paper empirically validates this process by providing PCA visualisation showing clusters split by languages.

Based on the decomposition framework, the intervention (i.e., making the representation language-agnositic), is applied at inference time on MGSM, XWinograd, M-MMLU on multiple LLM families. Results show consistent performance improvements across languages, LLM families, and datasets.

Furthermore, ablation4.1 shows that language-specific features (controlled by the amount to be subtracted in the intervention step) negatively correlate with reasoning accuracy. Ablation4.2 shows that the findings generally hold across layers (except the very last one). Ablation4.3 shows that the proposed intervention can provide performance improvements as much as post-training.

**Questions:**

Please refer to the weakness, especially weakness1. I believe the work (and its findings) can have more impact to the broader ML/NLP community if the invention can be integrated into post-training, allowing trained models to perform better at inference without any additional computation cost.

**Ethical Concerns:**

["NO or VERY MINOR ethics concerns only"]

**Final Justification:**

Given the strengths of the paper with the weaknesses -- computational overhead and potentially limited benefit in maths. The main contribution will lie in the insights & findings of the paper (e.g., time-time decomposition & intervention) -- the rebuttal provides further details about these two points; however, they can still be seen as limitations for the work considered as a methodology contribution. My original assessment is already leaning accept, so I'll keep this original assessment, but I raised its clarity

**Limitations:**

yes

**Quality:**

3

**Strengths And Weaknesses:**

Strengths
1. The paper provides useful insights related to multilingual reasoning, that language and reasoning could be decoupled.
2. The intervention method, originally serving as a probing mechanism, allows improved performance (as much as post-training)
3. The paper provides several ablation studies to strengthen the general finding about language and reasoning decomposition.

Weaknesses
1. The findings in this paper could have a limited impact. First, from the performance perspective, although the intervention can provide performance improvement, its additional computation could make it less practical. Second, it’d be better if the paper went deeper into suggesting how post-training could adopt this intervention into improving multilingual reasoning at the post-training stage (instead of having to apply it at inference time).
2. [minor point] I’m not sure if the datasets (e.g., MGSM) are too easy for current LLMs, or if the drop in performance in low-resource languages might actually come from poor machine translation (when the datasets were constructed)?

---

> ### Author Rebuttal · Authors · 2025-07-30
>
> Thank you for your thoughtful feedback. Below we address your concerns point by point.
>
> ---
>
> > **W1.1: its additional computation could make it less practical**
>
> Thank you for your feedback. Regarding the concern on computational overhead, we conducted a speed test on the **MGSM dataset** using **DeepSeek-R1-Distill-Qwen-7B** on a **single NVIDIA A100 GPU**.
>
> |Model|Inference speed (token/s)|
> |---|---|
> |Origin Model|2618.63|
> |Ours|2342.98|
>
> The results show that our method causes **minimal slowdown** in inference speed.
>
> Importantly, the intervention itself involves only a **lightweight projection and addition operation**, which is negligible compared to the total cost of running the LLM. The slight speed drop mainly comes from our **vLLM-based implementation**, not from the method itself.
>
> We are actively working on optimizing the code to further reduce overhead and bring the inference speed closer to the original model, enhancing the method’s practicality.
>
> ---
>
> > **W1.2: post-training integration**
>
> We have discussed the potential integration of our **language–reasoning decoupling** approach into training in **Appendix A (Lines 640–646)**. For example, using the ablated representation as a **supervision signal** could encourage **language-invariant intermediate representations** during supervised fine-tuning or reinforcement learning, potentially improving generalization.
>
> However, implementing and validating this at training time is beyond the scope of this paper and is left as future work.
>
> ---
>
> > **W2: Dataset difficulty and translation quality**
>
> Thank you for the thoughtful observation. We agree that some of the datasets used, such as MGSM, are relatively easy for modern LLMs.
>
> Translation quality is also a valid concern. To address this, we introduced **PolyMath** [1], a **human-verified multilingual dataset**. For each language, we sampled **50 questions from each of three difficulty levels (low, medium, high)** and computed a **difficulty-weighted score**.
>
> - DeepSeek-R1-Distill-Qwen-7B
>
> |Model|Bn|Sw|Th|Te|En|Es|Fr|De|Zh|Jp|Ru|Avg.|Average Fidelity|
> |---|---|---|---|---|---|---|---|---|---|---|---|---|---|
> |Origin Model|42.29|21.14|**42.29**|28.29|48.86|46.29|41.14|49.14|45.14|**45.71**|39.71|40.91|41.94|
> |Ours|**45.14**|**21.71**|41.71|**34.00**|**50.29**|**50.86**|**48.00**|**51.14**|**46.57**|43.43|**46.86**|**43.31**|**44.18**|
>
> - Qwen2.5-7b-Instruct
>
> |Model|Bn|Sw|Th|Te|En|Es|Fr|De|Zh|Jp|Ru|Avg.|Average Fidelity|
> |---|---|---|---|---|---|---|---|---|---|---|---|---|---|
> |Origin Model|17.14|9.43|16.57|14.29|23.14|23.14|22.29|**21.43**|18.29|20.00|28.86|19.51|42.00|
> |Ours|**18.57**|**16.29**|**21.43**|**17.14**|**25.71**|**28.86**|**25.14**|20.57|**22.00**|**23.43**|**28.00**|**22.47**|**42.00**|
>
> Experiments on DeepSeek-R1-Distill-Qwen-7B and Qwen-2.5-Instruct-7B show that, while higher-quality and balanced datasets reduce the performance gap between English and other high-resource languages (e.g., Spanish, French), **mid- and low-resource languages** like Swahili and Telugu still underperform—confirming that capability gaps persist.
>
> Our method helps **close this gap**, notably **elevating performance** in weaker languages (e.g., Russian in DeepSeek, Thai in Qwen) to levels comparable with stronger languages. This demonstrates the **practical value** and **effectiveness** of our intervention.
>
> Reference:
>
> [1] PolyMath: Evaluating Mathematical Reasoning in Multilingual Contexts.
>
> ---
>
> We sincerely appreciate your comments and will revise the manuscript to better reflect these clarifications.

---

> > ### Comment · Reviewer_EnSg · 2025-08-04
> >
> > Thank you for your detailed rebuttal, and for providing further justification weaknesses in (1) and (2). Given the strengths of the paper with the weaknesses -- computational overhead and potentially limited benefit on maths. The main contribution will lie in the insights & findings of the paper (e.g., time-time decomposition & intervention) and my original assessment is already leaning accept, so I'll keep this original assessment.

---

> > > ### Author Response · Authors · 2025-08-05
> > >
> > > Thank you very much for your kind response and for acknowledging the insights and contributions of our work. We’re glad to hear that our rebuttal addressed your concerns, and we sincerely appreciate your support and positive assessment of the paper.

---

### Official Review · Reviewer_iAev · 2025-06-27

**Clarity:** 3
**Significance:** 4
**Originality:** 4
**Rating:** 5
**Confidence:** 4

**Summary:**

The paper proposes a method to disentangle the language-agnostic reasoning capabilities of a multilingual LLM from its language-specific knowledge. They showcase how attenuating the language-specific signals enhance the reasoning performance of multiple LLMs across all the languages supported by the corresponding LLM. Their method, while being pretty lightweight, yields some impressive results comparable to post-training techniques like Reinforcement Learning (PPO).

**Questions:**

(Q1) In the ‘Implementation Details’ section the authors say "For each model, we ablate the language-specific components from mid-layer hidden states and then re-inject them at higher layers.” What specific layers are these higher layers?
(Q2) In figure 3, it looks like the inverse of the expectation happens in the vicinity of 0. For example, the increase in performance for DeepSeek-R1-Distill-Qwen-7Band QwQ when the ablation strength is -0.1, and the decrease in performance for Qwen2.5-7B-Instruct when the ablation strength is 0.1. What are the authors' intuitions for these?
(Q3) In Figure 4, ablation in the later layers leads to much sharper drops in reasoning fidelity, even if not the worst performance in the case of  DeepSeek-R1-Distill-Qwen-7Band QwQ.This appears to influence the author’s decision to reintroduce language-specific components at higher layers (as mentioned in the implementation section). A discussion on why preserving reasoning fidelity is important would be useful.

**Ethical Concerns:**

["NO or VERY MINOR ethics concerns only"]

**Final Justification:**

As stated in my review, this is a good paper and the authors have adequately addressed my concerns.

**Limitations:**

yes

**Quality:**

3

**Strengths And Weaknesses:**

## Strengths:
(S1) The paper is well written, and the method is simple and easy to integrate in a pre-existing deployed LLM with no training required.
(S2) Various types of LLMs like non-thinking Qwen, distilled reasoning models like R1-Distill-Qwen and reasoning models training using RL like QwQ are studied which covers the broader categories of language models.
(S3) Solid analysis across models and layers and good choice of metrics like reasoning fidelity, generation fidelity alongside performance. This helps paint a good picture of how and where the proposed method helps.

## Weaknesses:
(W1) The language choices are diverse but there are only two languages in the mid and low-resource categories so it’s tough to see the performance impact especially on these categories.
(W2) On page 7, the authors say “we observe that ablation applied at most layers, especially from lower to middle depths”. It is not clear how this method affects the performance when language is entangled with the knowledge we are measuring the performance on. For example, how does this affect the performance on INCLUDE (https://arxiv.org/abs/2411.19799) which measures the efficacy of LLMs on regional knowledge corresponding to each language’s home countries?

---

> ### Author Rebuttal · Authors · 2025-07-30
>
> Thank you for your thoughtful feedback. We address each of your concerns below.
>
> ---
>
> > **W1. Limited number of mid/low-resource languages**
>
> Thank you for your comment and suggestion. While we include **11 typologically diverse languages** covering high-, mid-, and low-resource cases, we acknowledge that the number of mid/low-resource languages is still limited. This is mainly due to the **lack of widely adopted multilingual benchmarks** with sufficient coverage of truly low-resource languages. In future work, we aim to incorporate more underrepresented languages, **once suitable benchmarks become available**.
>
> ---
>
> > **W2: Entanglement of language and knowledge (e.g., INCLUDE benchmark)**
>
> Thank you for raising this important point. To address this, we conduct additional experiments on the **Regionality** subset of the INCLUDE benchmark, using **Qwen2.5-7B-Instruct** and **DeepSeek-R1-Distill-Qwen-7B**. The results are summarized below:
>
> - Qwen2.5-7B-Instruct
>
> |Model|Bn|Te|Zh|Jp|Es|Fr|De|Ru|Avg.|Average Fidelity|
> |---|---|---|---|---|---|---|---|---|---|---|
> |Origin Model|49.40|35.34|67.81|71.06|65.00|63.07|**53.93**|55.98|57.70|11.57|
> |Ours|**51.00**|**36.14**|**70.02**|**75.65**|**67.40**|**67.12**|51.69|**60.56**|**59.95**|**11.66**|
>
> - DeepSeek-R1-Distill-Qwen-7B
>
> |Model|Bn|Te|Zh|Jp|Es|Fr|De|Ru|Avg.|Average Fidelity|
> |---|---|---|---|---|---|---|---|---|---|---|
> |Origin Model|34.54|29.72|50.30|56.09|54.60|49.87|43.82|**49.80**|46.09|12.37|
> |Ours|**41.37**|**32.13**|**51.11**|**57.09**|**56.00**|**53.10**|**51.69**|48.61|**48.89**|**12.41**|
>
> We find that even under strong language–knowledge entanglement, our method still yields **consistent improvements**, suggesting that it maintains **robust reasoning** while preserving **localized knowledge fidelity**. We will include this analysis and discussion in the revised version to highlight the **generality and practical value** of our approach.
>
> ---
>
> > **Q1: What are the higher layers used for re-injection?**
>
> Thank you for the question. The specific middle and higher layers used for ablation and re-injection vary by model. The selection is roughly based on dividing the model into three segments (lower, middle, higher), as inspired by recent interpretability studies. The exact layer ranges for each model are as follows:
>
> |Model Name|Number of Layers|Middle Layers|Higher Layers|
> |---|---|---|---|
> |Qwen-2.5-Instruct-3B|36|12-26|27-35|
> |Qwen-2.5-Instruct-7B|28|10-19|20-27|
> |Qwen-3-1.7B-Thinking|28|10-19|20-27|
> |Qwen-3-4B-Thinking|36|12-26|27-35|
> |Qwen-3-8B-Thinking|36|12-26|27-35|
> |R1-Distill-Qwen-7B|28|10-19|20-27|
> |R1-Distill-LLama-8B|32|12-22|23-31|
> |R1-Distill-Qwen-14B|48|16-33|34-47|
> |GLM-Z1-9B|40|12-30|31-39|
> |QwQ-32B|64|20-46|47-63|
>
> ---
>
> > **Q2: Unexpected effects near ablation strength 0**
>
> Indeed, some models exhibit **minor deviations from the overall trend** when the ablation strength is near 0. Upon further analysis, we found that this is primarily due to **Japanese and Thai**, which behave anomalously in both performance and language fidelity in this narrow region.
>
> We believe this is related to the **surface-form similarity** between Japanese/Thai and Chinese.
> As illustrated in **Figures 4 (a)–9 (a)**, the lower-layer representations of Japanese and Thai are quite similar, and as shown in **Figures 4 (c)–9 (c)**, at higher layers, both languages become **closer to Chinese** than to English. In practice, we observed that when slight ablation is applied, the model may even **respond in Chinese to Japanese or Thai prompts**, in contrast to its usual tendency to **reply in English to prompts in other languages**.
>
> Our intuition is that these **“neighboring” languages** do not form fully independent anchors in the multilingual semantic space. As a result, **small changes in ablation strength may trigger abrupt shifts in representation**, leading to local irregularities in both reasoning and language output.
>
> This highlights the **complex internal dynamics of multilingual LLMs**, particularly for **lower-resource languages that are structurally close to high-resource ones**. We plan to investigate this further in future work by analyzing language clustering in representation space and exploring **language-specific or adaptive interventions** to improve stability and fairness.
>
> ---
>
> > **Q3: Why is preserving reasoning fidelity important?**
>
> Thank you for the question. We believe that preserving language fidelity is crucial for multilingual LLMs.
>
> On one hand, it ensures that users can interact with the model **fluently in their native language**, which is essential for usability and accessibility. On the other hand, language consistency reflects the model’s **ability to understand and maintain context**, supporting accurate and coherent information exchange.
>
> Therefore, we view maintaining **alignment between input and output languages** as a necessary goal for practical multilingual applications.
>
> ---
>
> Thank you again for your valuable feedback and we will incorporate your suggestions to improve the paper.

---

> > ### Comment · Reviewer_iAev · 2025-08-04
> >
> > Thank you for your response. I've updated my score to a 5.

---

> > > ### Author Response · Authors · 2025-08-05
> > >
> > > Thank you very much for your updated score and for your thoughtful feedback throughout the review process. We truly appreciate your time and support!

---

### Official Review · Reviewer_RfPJ · 2025-06-30

**Clarity:** 3
**Significance:** 3
**Originality:** 3
**Rating:** 5
**Confidence:** 4

**Summary:**

The work uses a method to disentangle language specific and reasoning processing in language models. This is done by first constructing mean embedding matrix M for multilingual latent space (using embeddings of the final token triggered by samples of all available languages) and computing via singular value decomposition (SVD) the language specific subspace M_s (based on formulation of M introduced in previous works), which is used to project out components of hidden activations h from any network layer. This is supposed to remove language specific variations and to ensure remaining representation \hat{h} reflects only language agnostic reasoning relevant content. Authors perform then interventions on activation space on a diverse set of language and reasoning models. For the interventions, they remove language specific components from activations in middle layers, while plugging those back in upper layers. Doing so, authors observe improvements on broad range of language understanding and reasoning benchmarks (MGSM, XWinograd, M-MMLU), across multiple languages. Further, they compare for Qwen 2.5 Instruct 3B the training free intervention with SFT and RL posttraining, observing in majority of the cases activation intervention being as effective or even outperforming posttraining. Authors conclude that their training free method to dampen language specific activations in mid layers while keeping those intact in top layers is significantly improving problem solving performance and argue for benefits of disentanglement of core reasoning and language specific processing.

**Questions:**

Following the weaknesses section,

1. How is language generation affected by the intervention, is there any way to check whether it is still intact or breaking down?
2. Can we be sure that intervention based on M_s is special and eg even random projection to subspace of same rank would not give similar positive effects?
3. Is there any conclusion from observations done in the study on how to improve pre-training in general using language-specific and language-agnostic reasoning disentanglement?
4. Is the benefit of training free vs SFT/RL posttraining limited to smaller scale 3B only? Will it hold eg for 7B as well?

**Ethical Concerns:**

["NO or VERY MINOR ethics concerns only"]

**Final Justification:**

Authors provided extensive experiments per request that substantially strengthened the work. I increase therefore the score accordingly from initial 3 to 5.

**Limitations:**

Following the weaknesses section -
1. No conclusion can be made for effect of intervention on language generation capability. UPDATE: experiment done
2. Claimed benefit of training free intervention vs SFT/RL posttraining is limited to one small scale only (3B)

**Paper Formatting Concerns:**

No major issues.

**Quality:**

4

**Strengths And Weaknesses:**

The strengths of the study:

1. Training free method that according to evaluations results in significant improvement on number of language understanding and reasoning benchmarks across various high, mid and low resource languages
2. Thorough experiments looking at the effect of activation interventions across various layers, allowing for detailed conclusion about differential effect of the intervention on the middle and top layers.
3. Evaluations done on variety of relevant models.
4. Important for dealing with multi-lingual setting, offers interesting angle on language specific and language agnostic function.


The weaknesses:
1. It is not clear how ablations affect the language generation quality in multi lingual setting. Authors show improvements on problem solving reasoning benchmarks. Those can be entirely decorellated from language generation capability, eg model can solve reasoning tasks posed in language X but entirely fail to generate fluent text in the same language. As authors argue for benefits in multi lingual setting, I think it is important to understand how language generation is affected by the intervention (eg using evals like ROUGE, BLEU, METEOR, etc)
2. Control for the causal effect of the intervention is missing. It can be that other various projections into subspace of same rank would affect the reasoning quality. As authors explicitly argue for causal effect, control eg, comparison of authors methods to projection on random subspace of same rank with corresponding activation interventions based on that, is required to check this claim.
3. Authors do not offer any hints or experiments for improving training. If it is true that disentanglement of language specific and language agnostic reasoning core is beneficial for reasoning function, while retaining language comprehension and generation capabilities, it would be important to provide at least some thoughts on how pretraining can be shaped to improve the model function accordingly (even better would be of course experiment showing that it works)
4. The comparison to SFT and RL is done only for rather small scale 3B model. It is well known that larger scale models benefit more from posttraining. It would be insightful to see how training free intervention compares to post training on larger scales, eg at least 7B. Claims that training free intervention matches or outperforms posttraining are in current form limited to 3B scale only and claim should be correspondingly toned down.
5. Authors repeatedly refer to open-weights models as open-source models. This is not correct. Open source models allow full study and modification of their origins including pre-training procedure, for which fully open and available data is required. Many of the models referred by authors, eg Llama or Qwen families, do not have open data, and are thus not open-source, but only open weights.
6. The source code offered by authors in supplementary for reproduction is hardly documented, it is not straightforward to repeat experiments in the current form as described in the paper.

---

> ### Author Rebuttal · Authors · 2025-07-30
>
> Thank you for your thoughtful and detailed review. We appreciate your feedback and address each point below.
>
> ---
>
> > **W1: Impact on multilingual language generation quality**
>
> Thank you for the suggestion. To address your concern, we conducted additional **machine translation** experiments by translating English sentences into target languages, in order to assess the impact of our method on language generation quality.
>
> Specifically, we used the **Flores-101** dataset and sampled **200 sentences per language**. We evaluated **ROUGE** and **BLEU** scores on **Qwen2.5-7B-Instruct** and **DeepSeek-R1-Distill-Qwen-7B**.
>
> - DeepSeek-R1-Distill-Qwen-7B
>
> |Model|ROUGE-1|ROUGE-2|ROUGE-L|BLEU|
> |---|---|---|---|---|
> |Origin Model|26.80|10.15|24.17|8.63|
> |Ours|**27.74**|**10.67**|**25.30**|**9.07**|
>
> - Qwen2.5-7b-Instruct
>
> |Model|ROUGE-1|ROUGE-2|ROUGE-L|BLEU|
> |---|---|---|---|---|
> |Origin Model|38.21|19.82|35.36|17.00|
> |Ours|**38.82**|**20.18**|**36.03**|**17.43**|
>
> Results show that our method does **not degrade language generation quality** in the prompted language. We will include these results in the revision.
>
> ---
>
> > **W2: Causal effect of projection intervention**
>
> Thank you for this important point. We fully agree that without a **random subspace control**, it is difficult to establish the **specificity** of the language subspace used in our method.
>
> To address this, we generated **random subspaces of the same rank** as our extracted multilingual subspace and replaced the original subspace in the intervention. We then evaluated the performance on **MGSM** using **Qwen2.5-7B-Instruct** and **DeepSeek-R1-Distill-Qwen-7B**.
>
> - DeepSeek-R1-Distill-Qwen-7B
>
> |Model|Performance|Fidelity|
> |---|---|---|
> |Origin Model|56.11|90.98|
> |Ours|**58.51**|**92.18**|
> | w/ random subspace|55.27|89.67|
>
> - Qwen2.5-7b-Instruct
>
> |Model|Performance|Fidelity|
> |---|---|---|
> |Origin Model|69.82|79.75|
> |Ours|**70.76**|**84.44**|
> |w/ random subspace|68.65|79.20|
>
> The results show that with random subspaces, both **performance and fidelity degrade**, whereas our method yields **consistent improvements**. This supports a **causal link** between the extracted language subspace and the observed reasoning gains. We will include these control results in the revision.
>
> ---
>
> > **W3: Implications for training**
>
> Thank you for the suggestion.
>
> We have discussed the potential integration of our **language–reasoning decoupling** approach into training in **Appendix A (Lines 640–646)**. For example, using the ablated representation as a **supervision signal** could encourage **language-invariant intermediate representations** during supervised fine-tuning or reinforcement learning, potentially improving generalization.
>
> However, implementing and validating this at training time is beyond the scope of this paper and is left as future work.
>
> ---
>
> > **W4: Limited scale of post-training comparisons**
>
> Thank you for the suggestion. We agree that comparing against post-training methods at larger scales would strengthen the paper. Due to **limited computational resources**, we were unable to run such large-scale post-training baselines.
>
> However, as shown in **Tables 1 and 2**, our **training-free intervention consistently improves multilingual reasoning** across a wide range of model sizes, from **1.7B to 32B**. These results demonstrate the scalability and effectiveness of our method, even without additional training.
>
> ---
>
> > **W5: Clarification on open-source terminology**
>
> Thank you for pointing this out. We acknowledge the misuse of the term “open-source models.” In the revision, we have corrected all instances to “open-weight models” to accurately reflect that these models only release weights, not training data or pretraining details.
>
> ---
>
> > **W6: Reproducibility of code**
>
> Thank you for the feedback. We apologize for the incomplete documentation.
>
> The released code covers all core components, including **vLLM-based intervention**, **language subspace extraction**, and **main evaluation scripts**, and supports reproducing key results.
>
> We are committed to improving reproducibility and will release a fully documented version shortly. Thank you again—we appreciate the suggestion and will address it promptly.
>
> ---
>
> For each of the question you raised, we have provided point-by-point responses above. Please kindly refer to our replies to Weaknesses 1–6 for detailed clarifications and corresponding experimental or revision plans.
>
> We truly appreciate your constructive feedback and hope our responses address your concerns.

---

> > ### Comment · Reviewer_RfPJ · 2025-08-02
> >
> > I thank the authors for the extensive reply and for additional experiments that help to clarify my questions. It is insightful to see that language generation is not compromised and seems to actually even improve alongside reasoning. Also having control experiment with random subspace of the same rank leading to drop in performance is reassuring for the original method to be the cause for the observed improvements.
> >
> > In light of these additional experiments and explanations, I am considering to increase the score by 1 point.
> >
> > What makes me still somewhat hesitant is the choice of main reasoning benchmarks. I think the benchmarks used for testing reasoning capabilities are lacking widespread ones like AIME24/25, MATH500 and GPQA/GPQA-Diamond that are used for comparing reasoning models in most of the works. It would be good to see whether the intervention does not hurt the performance on those benchmarks or even increases the scores. Although these benchs are english only and do not offer multi lingual scenario, they are important reference for comparing reasoning capability and for the claim that intervention is boosting reasoning to be strong, it is necessary to check performance there beside multi lingual benchs the authors are using.
> >
> > Further, it would be good to understand how sensitive the construction of the subspace is to choice of samples from each language that leads to composition of the mean embeddings matrix. Authors should elaborate in more details how they chose samples to construct mean embedding matrix and discuss whether the method is sensitive to the particular sample choice (ideally, providing an experiment that measures sensitivity or lack thereof). This can also motivate scoring up.

---

> ### Author Response · Authors · 2025-08-03
>
> Thank you very much for your thoughtful reply and your willingness to consider increasing the score. We truly appreciate your continued engagement. Below, we address the remaining concerns with new experiments and clarifications.
>
> ---
>
> > **1. Evaluation on standard reasoning benchmarks (AIME24/25, MATH500, GPQA-Diamond)**
>
> We fully agree that evaluating on **widely-used English reasoning benchmarks** is important to support the general claim that our intervention improves reasoning capabilities.
>
> To this end, we conducted additional experiments using **DeepSeek-R1-Distill-Qwen-7B**, applying our intervention on **AIME 2024, AIME 2025, MATH500, and GPQA-Diamond**. Results are shown below:
>
> | Model  | AIME24  | AIME25  | Math500 | GPQA_Diamond |
> |--------|---------|---------|---------|-----------|
> | Origin | 48.33   | 35.83   | 92.00   | **48.99** |
> | + Ours   | **52.92** | **45.83** | **92.80** | 47.98     |
>
> We observe that our method consistently improves performance even on these English-only benchmarks, further supporting our core claim:
>
> *Reasoning and language processing can be explicitly decoupled, and reasoning mechanisms in LLMs are largely language-agnostic.*
>
> The reason for improved performance in English, is that the language-agnostic representation space is often anchored and implicitly aligned to English, due to its dominance in training data. Our intervention encourages models to operate more directly in this aligned space, benefiting even English reasoning tasks.
>
> We also extend our evaluation using PolyMath, PolyMath [1], a human-verified multilingual dataset. For each language, we sampled 50 questions from each of three difficulty levels (low, medium, high) and computed a difficulty-weighted score. Results again show consistent improvements across difficulty levels and languages, reinforcing the robustness and effectiveness of our approach.
>
> - DeepSeek-R1-Distill-Qwen-7B
>
> |Model|Bn|Sw|Th|Te|En|Es|Fr|De|Zh|Jp|Ru|Avg.|Average Fidelity|
> |---|---|---|---|---|---|---|---|---|---|---|---|---|---|
> |Origin Model|42.29|21.14|**42.29**|28.29|48.86|46.29|41.14|49.14|45.14|**45.71**|39.71|40.91|41.94|
> |Ours|**45.14**|**21.71**|41.71|**34.00**|**50.29**|**50.86**|**48.00**|**51.14**|**46.57**|43.43|**46.86**|**43.31**|**44.18**|
>
> Reference:
>
> [1] PolyMath: Evaluating Mathematical Reasoning in Multilingual Contexts.
>
> ---
>
> > **2. Sensitivity of the mean embedding matrix to sample choice**
>
> Thank you for raising this important point. In our current setup (Appendix F, Lines 743–746), we construct the mean embedding matrix using the **7,500 training examples from MATH**, translated into 10 languages via **Google Translate**. These samples were **not handpicked**, and we **did not manually verify** translation quality, suggesting that our method is already **reasonably robust** to sample noise and translation imperfections.
>
> To directly test **sensitivity**, we conducted additional experiments by varying the **number of samples** used to construct the embedding matrix. We tested 5%, 20% and 50% examples.
>
> | Data Used (%) | Performance | Fidelity |
> |:-------------:|:-----------:|:--------:|
> | Origin Model  | 56.11       | 90.98    |
> | 5%            | 57.36       | 91.68    |
> | 20%           | 58.02       | 91.56    |
> | 50%           | **58.56**       | 91.16    |
> | 100%          | 58.51   | **92.18** |
>
> The results show that performance remains **relatively stable across different sample sizes**, indicating that our approach is **not highly sensitive to the exact sample set**, and the constructed language subspace is **robust and generalizable**. We will include this discussion and the supporting results in the revised paper.
>
> ---
>
> We thank you again for the thoughtful feedback—your suggestions have greatly strengthened our paper, both in scope and in evidence.

---

> > ### Comment · Reviewer_RfPJ · 2025-08-03
> >
> > Thanks for the extensive experiments, I think it strengthens the paper substantially. I will reflect it correspondingly in the scores.

---

> > > ### Author Response · Authors · 2025-08-04
> > >
> > > Thank you very much for your positive feedback and for considering an increased score. We truly appreciate your thoughtful comments and engagement throughout the review process—it has helped us significantly improve the quality and clarity of our work.

---

### Official Review · Reviewer_trBL · 2025-07-01

**Clarity:** 4
**Significance:** 3
**Originality:** 3
**Rating:** 5
**Confidence:** 4

**Summary:**

This paper proposes a method to improve multilingual reasoning in large language models (LLMs) by disentangling language and reasoning representations. The authors hypothesize that, similar to the human brain, language and reasoning are separable components in LLMs. They introduce a technique to ablate language-specific information from the model's hidden states during inference. Their experiments, conducted on 10 open-source LLMs across 11 languages and 3 benchmarks, show that this intervention consistently improves multilingual reasoning performance. The paper also provides an analysis of the effects of this ablation, showing that it encourages the model to rely on a more language-agnostic representation for reasoning, with English often serving as an anchor.

**Questions:**

1.  The paper's results seem to depend heavily on the choice of layers for ablation. Could you clarify which specific layers were selected for the results presented in Tables 1, 2, and 3, and what was the rationale for this choice?

2.  Was language re-injected (decoupling) for the results in the main tables to achieve the high language fidelity, or was it pure ablation? If it was re-injected, could you provide the implementation details (e.g., which layers, what rule was used, similar to Equation 3)?

3. For a given prompt language, maybe it would be more relevant to only ablate/suppress (idiosyncrasies of) *other* languages and not the prompted one. Maybe this would allow to strongly ablate to get better reasoning accuracy while preserving language fidelity (without re-injecting/decoupling).

**Ethical Concerns:**

["NO or VERY MINOR ethics concerns only"]

**Final Justification:**

The authors' clarifications have successfully addressed all of my concerns.

The explanation of the distinction between ablation and decoupling, along with the provided table detailing the specific layers used for each model, resolves the main ambiguities in the methodology. Including this information in the revised manuscript will significantly improve the paper's clarity and reproducibility.

Given these comprehensive revisions, I am confident that the paper will be much stronger. I revised the scores accordingly.

**Limitations:**

Yes

**Quality:**

4

**Strengths And Weaknesses:**

## Strengths

- The paper addresses the important and challenging problem of improving multilingual reasoning in LLMs.
- The proposed method is intuitive, training-free, and shows promising results on a wide range of models and languages.
- The paper includes a good amount of analysis to understand the effect of the proposed intervention.

## Weaknesses

### Unclear methodology

- The difference between *ablation* and *decoupling* seems very important and is unclear in the paper as is.
- As far as I understand, ablation = suppressing language in early/mid layers and decoupling = ablation + re-injecting language in later layers.
- Apart from Figure 4 which is explicitly about decoupling, it seems that all results are about ablation, is it really the case?
- There are no implementation details on the re-injection. Which layer did you choose and why? Do you use a similar rule as (3)?

### A claim is not supported

Here I will assume that the dashed lines on Figure 3 and 4 represent the performance of base models.

From lines 230-233: "As shown in Figure 4, we observe that ablation applied at most layers, especially from lower to middle depths, consistently improves reasoning accuracy while maintaining stable output fidelity. This suggests that language–reasoning decoupling is broadly effective across the network and does not require precise targeting to specific layers to yield benefits."

- While it is true that reasoning and response language fidelity remain high, only QwQ actually benefits from the decoupling in terms of reasoning accuracy, while Qwen2.5's accuracy is even impaired.
- Figure 3 and 4 show that there is a trade-off and that it seems not possible to improve reasoning without impairing language with this decoupling method. Therefore not successfully decoupling language and reasoning.
- This makes me wonder:
   * Which ablation layer has been chosen for the results in the different tables?
   * Has language been re-injected (decoupling instead of ablation) to ensure the high language fidelity displayed?

### Poor visualization of results

The main tables (1, 2, and 3) are very cumbersome to read. They would be easier to parse and the global effect of the ablation would be clearer if the results were presented as barplots or lineplots (e.g., Table 1 could be a series of barplots, one for each language, with models on the x-axis and performance on the y-axis, using shading to distinguish base vs. ablated models). Additionally, the captions for Figures 3 and 4 are incomplete, as they do not define the dashed lines, which presumably represent the baseline performance.


### Misc
The paper would be stronger if the limitation section from the appendix was moved to the main body. Furthermore, the paper's checklist incorrectly claims that statistical significance is reported. The checklist states, "Statistical significance of the experiments can be found in Appendix F," which is not true; Appendix F contains no such information.

---

> ### Author Rebuttal · Authors · 2025-07-30
>
> Thank you for your thoughtful and constructive review. We sincerely appreciate your feedback and address each of your concerns point by point below.
>
> ---
>
> > **W1: Unclear methodology**
>
> Thank you for the valuable feedback, and we apologize for the confusion.
>
> Your understanding is correct: ablation suppresses language signals in mid layers, while decoupling further re-injects them in higher layers. All reported main experimental results are based on decoupling, as stated in Implementation Details (Lines 172–173).
>
> Regarding re-injection, we use a similar layer-wise rule as in Equation (3), which is used for ablation; the only difference is that we add back the language component instead of subtracting it. The exact layers for ablation and re-injection per model are listed below:
>
> | Model Name                | Number of Layers | Ablation Layers | Re-Injection Layers |
> |---------------------------|------------------|---------------|---------------|
> | Qwen-2.5-Instruct-3B      | 36               | 12-26         | 27-35         |
> | Qwen-2.5-Instruct-7B      | 28               | 10-19         | 20-27         |
> | Qwen-3-1.7B-Thinking       | 28               | 10-19         | 20-27         |
> | Qwen-3-4B-Thinking         | 36               | 12-26         | 27-35         |
> | Qwen-3-8B-Thinking         | 36               | 12-26         | 27-35         |
> | R1-Distill-Qwen-7B        | 28               | 10-19         | 20-27         |
> | R1-Distill-LLama-8B       | 32               | 12-22         | 23-31         |
> | R1-Distill-Qwen-14B       | 48               | 16-33         | 34-47         |
> | GLM-Z1-9B                 | 40               | 12-30         | 31-39         |
> | QwQ-32B                   | 64               | 20-46         | 47-63         |
>
> ---
>
> > **W2: Unsupported claim**
>
> Thank you for your detailed comments. We would like to clarify the experimental settings and improve our descriptions accordingly.
>
> In our experiments, we roughly divide the model layers into lower, middle, and higher layers.
> - In **Figure 3**, we apply **ablation only in the middle layers**.
> - In **Figure 4**, we apply **ablation only on small groups of consecutive layers**.
> - In **Table 1 and Table 2**, we apply **ablation in the middle layers followed by re-injection in the higher layers**—i.e., full **decoupling**.
> The exact ablation and re-injection layers for each model are listed in our response to W1.
>
> As you correctly observed in **Figure 4**, ablating different layers does not always improve reasoning accuracy. Our wording here was not sufficiently rigorous. Compared to language fidelity, ablation at certain isolated layers can have unstable effects on reasoning. While **Figure 3** shows that middle-layer ablation consistently boosts reasoning, **Figure 4** focuses on the impact of ablation on **language fidelity**—especially the sharp drop when ablation is applied in **higher layers**.
>
> This is precisely why our main setup performs **middle-layer ablation and top-layer re-injection**: to improve reasoning while preserving output fluency and consistency.
>
> We fully agree with your point about the **reasoning–fidelity trade-off**. This trade-off is indeed present under ablation alone, and motivates our decoupling strategy.
>
> Thank you again for the constructive feedback. We will revise the manuscript to make these distinctions and justifications clearer.
>
> ---
>
> > **W3: Poor visualization of results**
>
> Thank you for the helpful suggestion.
>
> We fully agree that the main tables could be made more readable and the overall effect clearer through visualizations. In the revision, we will include bar/line plots for Tables 1–3 and improve the figure captions by explicitly defining the dashed lines (which indicate baseline performance).
>
> We appreciate your feedback and will revise accordingly.
>
> ---
>
> > **W4: Misc**
>
> We agree that moving the Limitation section into the main body will improve clarity and completeness, and we will make this change in the revision.
>
> Regarding the checklist: you are right—Appendix F does not currently include statistical significance results. This was an oversight on our part. We have conducted significance tests on the multilingual results, showing that our method yields statistically significant gains (p < 0.05) on most benchmarks. We believe this further supports the effectiveness of our approach, especially given its low cost and wide language coverage.
>
> We will include the statistical analysis and correct the checklist in the revised version. Thank you again for your careful review.
>
> ---
>
> > **Q1: choice of layers for ablation**
>
> For Tables 1, 2, and 3, we apply **ablation in the middle layers** and **re-injection in the higher layers**. The specific layer ranges used for each model are provided in our response to W1. This choice is inspired by recent interpretability studies on multilingual representations in LLMs (see Lines 241–242).
>
> We will include this information directly in the revised paper for clarity.
>
> ---
>
> > **Q2: Regarding Re-Injection**
>
> Yes, language was re-injected in the higher layers for all main results—i.e., we used **decoupling**, not pure ablation. The specific re-injection layers for each model are listed in our response to W1.
>
> The implementation is nearly identical to ablation, except that we **add** the language component back instead of subtracting it.
>
> ---
>
> > **Q3: only ablate/suppress (idiosyncrasies of) other languages**
>
> Thank you for the insightful suggestion.
>
> We have tried using a language subspace that excludes the prompt language. However, in this case, the prompt has very little projection onto the subspace, so the ablation has minimal effect—leading to negligible changes in reasoning performance.
>
> We agree this is an interesting direction and worth further exploration.
>
> ---
>
> Once again, thank you for your constructive feedback. We hope our response has addressed your concern.

---

> > ### Comment · Reviewer_trBL · 2025-08-08
> >
> > Thank you for your detailed and thoughtful rebuttal. Your clarifications have successfully addressed all of my concerns.
> >
> > The explanation of the distinction between ablation and decoupling, along with the provided table detailing the specific layers used for each model, resolves the main ambiguities in the methodology. Including this information in the revised manuscript will significantly improve the paper's clarity and reproducibility.
> >
> > Given these comprehensive revisions, I am confident that the paper will be much stronger. I will revise my score accordingly.
> > Thank you for your thorough response.

---

> > > ### Author Response · Authors · 2025-08-08
> > >
> > > Thank you very much for your kind and encouraging response. We’re truly glad to hear that our clarifications have addressed your concerns, and we deeply appreciate your willingness to revise your score.
> > >
> > > Your thoughtful suggestions have played an important role in improving the clarity and completeness of our paper. Thank you again for your valuable feedback and support!

---

> ### Author Response · Authors · 2025-08-07
> **Kind Reminder to Reviewer trBL**
>
> Dear Reviewer trBL,
>
> Thank you again for your thoughtful feedback and valuable comments on our paper. We just wanted to kindly follow up and check if our rebuttal and the additional experiments have sufficiently addressed your concerns.
>
> If there are any remaining questions or clarifications we can provide, we’d be more than happy to assist.
>
> We truly appreciate your time and consideration.

---

### Decision · Program_Chairs · 2025-09-17

**Decision:**

Accept (spotlight)

**Comment:**

This paper describes interventions that can be applied to improve the performance of LLMs by decoupling the aspects that are responsible for the details of language encoding from those concerned with logical reasoning.

Strengths:

The reviewers appreciated the fact that the method did not involve extra training and could be applied to a wide range of existing models as shown in the provided results.

In the rebuttal the authors were able to address concerns regarding computational effort with measurements. They also clarified how they chose layers in the model for ablation and re-injection.

The authors also successfully addressed the concerns raised by reviewer  RfPJ regarding evaluating the disfluencies that could be introduced by the intervention in the language generation phase.


All of the reviewers were favorably inclined towards this submission and felt that it represented a useful contribution to improving LLM performance by decoupling language interpretation from reasoning in a way that would allow lower resourced languages to take more advantage of the reasoning power gleaned from higher resourced languages.